# A plasma membrane-localized polycystin-1/polycystin-2 complex in endothelial cells elicits vasodilation

Charles E MacKay, Miranda Floen, M Dennis Leo, Raquibul Hasan, Tessa AC Garrud, Carlos Fernández-Peña, Purnima Singh, Kafait U Malik, Jonathan H Jaggar*

Department of Physiology, University of Tennessee Health Science Center, Memphis, United States

*For correspondence: jjaggar@uthsc.edu

Competing interest: The authors declare that no competing interests exist.

**Abstract** Polycystin-1 (PC-1, PKD1), a receptor-like protein expressed by the *Pkd1* gene, is present in a wide variety of cell types, but its cellular location, signaling mechanisms, and physiological functions are poorly understood. Here, by studying tamoxifen-inducible, endothelial cell (EC)-specific *Pkd1* knockout (*Pkd1* ecKO) mice, we show that flow activates PC-1-mediated, $Ca^{2+}$-dependent cation currents in ECs. EC-specific PC-1 knockout attenuates flow-mediated arterial hyperpolarization and vasodilation. PC-1-dependent vasodilation occurs over the entire functional shear stress range and via the activation of endothelial nitric oxide synthase (eNOS) and intermediate (IK)- and small (SK)-conductance $Ca^{2+}$-activated $K^+$ channels. EC-specific PC-1 knockout increases systemic blood pressure without altering kidney anatomy. PC-1 coimmunoprecipitates with polycystin-2 (PC-2, PKD2), a TRP polycystin channel, and clusters of both proteins locate in nanoscale proximity in the EC plasma membrane. Knockout of either PC-1 or PC-2 (*Pkd2* ecKO mice) abolishes surface clusters of both PC-1 and PC-2 in ECs. Single knockout of PC-1 or PC-2 or double knockout of PC-1 and PC-2 (*Pkd1*/*Pkd2* ecKO mice) similarly attenuates flow-mediated vasodilation. Flow stimulates nonselective cation currents in ECs that are similarly inhibited by either PC-1 or PC-2 knockout or by interference peptides corresponding to the C-terminus coiled-coil domains present in PC-1 or PC-2. In summary, we show that PC-1 regulates arterial contractility through the formation of an interdependent signaling complex with PC-2 in ECs. Flow stimulates PC-1/PC-2 clusters in the EC plasma membrane, leading to eNOS, IK channel, and SK channel activation, vasodilation, and a reduction in blood pressure.

## Editor's evaluation

This is a very significant study that enhances our understanding of a mysterious ion channel and its function in vascular function.

## Introduction

Blood vessels are lined by endothelial cells (ECs), which regulate several physiological functions, including contractility, to control regional organ flow and systemic pressure. ECs can release several diffusible factors, including nitric oxide (NO), which relaxes arterial smooth muscle cells, leading to vasodilation (*Vane, 1993*). ECs also electrically couple to smooth muscle cells and directly modulate their membrane potential to modify arterial contractility (*Garland et al., 2011*). Several receptor agonists, substances, and mechanical stimuli, such as intravascular flow, are known to act in an EC-dependent

manner to regulate arterial functions. In many cases, the molecular mechanisms by which these physiological stimuli activate signaling in ECs to modulate vascular contractility are unclear.

Polycystin-1 (PC-1, PKD1) is a receptor-like transmembrane protein encoded by the *Pkd1* gene (*Hughes et al., 1995*). PC-1 is expressed in various cell types, including ECs, and is predicted to form eleven transmembrane helices, an extracellular N-terminus, and an intracellular C-terminus (*Hughes et al., 1995*; *Boulter et al., 2001*; *Bulley et al., 2018*; *MacKay et al., 2020*; *Burn et al., 1995*; *Qian et al., 2002*). The PC-1 N-terminus is large (>3000 amino acid residues) and contains multiple putative adhesion- and ligand-binding sites (*Hughes et al., 1995*; *Burn et al., 1995*; *Qian et al., 2002*; *Hardy and Tsiokas, 2020*; *Zhou, 2009*; *Nauli et al., 2003*). As such, PC-1 is proposed to act as a mechanical sensor and ligand-receptor, although stimuli that activate PC-1 and its functional significance are unclear.

ECs also express polycystin-2 (PC-2, PKD2), a protein encoded by the *Pkd2* gene (*MacKay et al., 2020*). PC-2 is a member of the transient receptor potential (TRP) channel family and is also termed TRP polycystin 1 (TRPP1) (*Earley and Brayden, 2015*). PC-1 and PC-2 have been proposed to signal through independent and interdependent mechanisms, with much of this knowledge derived from experiments studying recombinant proteins and cultured cells (*Hardy and Tsiokas, 2020*; *Brill and Ehrlich, 2020*). Supporting their independence, PC-1 and PC-2 exhibit distinct developmental and expression profiles in different kidney cell types (*Foggensteiner et al., 2000*). PC-2 channels do not require PC-1 to traffic to primary cilia and generate currents in primary-cultured kidney collecting duct cells (*Liu et al., 2018*; *Arif Pavel et al., 2016*). PC-1 is also proposed to act as an atypical G protein-coupled receptor (*Parnell et al., 1998*). In addition, C- and N-terminus-deficient PC-2 can alone form a homotetrameric ion channel with each subunit containing six transmembrane domains, when visualized using cryo-EM (*Shen et al., 2016*; *Grieben et al., 2017*). Evidence supporting PC-1 and PC-2 interdependency includes that mutations in either *Pkd1* or *Pkd2* result in autosomal dominant polycystic kidney disease (ADPKD), the most prevalent monogenic disorder in humans (*Rossetti et al., 2007*). ADPKD is typically characterized by the appearance of renal cysts, but patients can develop hypertension before any kidney dysfunction and cardiovascular disease is the leading (~50%) cause of death in patients (*Valero et al., 1999*; *Martinez-Vea et al., 2004*; *Torres et al., 2007*; *Gabow, 1990*; *Bergmann et al., 2018*). Experiments studying recombinant proteins and cultured kidney cell lines have provided evidence that PC-1 and PC-2 can exist in a protein complex (*Nauli et al., 2003*; *Su et al., 2018*; *Qian et al., 1997*; *Newby et al., 2002*; *Zhu et al., 2011*; *Yu et al., 2009*; *Hanaoka et al., 2000*; *Delmas et al., 2004*). Several domains in PC-1 and PC-2 may physically interact, including their C-terminus coiled-coils (*Su et al., 2018*; *Qian et al., 1997*; *Zhu et al., 2011*; *Yu et al., 2009*; *Tsiokas et al., 1997*). The structure of a PC-1/PC-2 heterotetrameric complex, which forms in a 1:3 stoichiometry, has also been resolved using cryo-EM (*Su et al., 2018*). Despite more than two decades of research, signaling mechanisms and physiological functions of PC-1 and PC-2 and their potential dependency or independence are poorly understood, particularly in extra-renal cell types, such as ECs.

Here, we generated inducible, cell-specific PC-1 knockout (*Pkd1* ecKO) mice to investigate signaling mechanisms and physiological functions of PC-1 in ECs of resistance-size arteries. We also studied EC-specific PC-2 knockout (*Pkd2* ecKO) mice and produced PC-1/PC-2 double knockout (*Pkd1/Pkd2* ecKO) mice to investigate whether PC-1 acts in an independent manner or is dependent on PC-2 to respond to physiological stimuli and elicit functional responses. Our data demonstrate that PC-1 and PC-2 protein clusters form an interdependent plasma membrane complex in ECs which is activated by flow to produce vasodilation and reduce blood pressure.

## Results

### Generation and validation of tamoxifen-inducible, EC-specific *Pkd1* knockout mice

Mice with loxP sites flanking exons 2–4 of the *Pkd1* gene (*Pkd1^{fl/fl}*) were bred with tamoxifen-inducible EC-specific Cre mice (*Cdh5(PAC)-CreERT2*) to generate *Pkd1^{fl/fl}:Cdh5(PAC)-CreERT2* (*Pkd1* ecKO) mice. Genomic PCR indicated that tamoxifen (i.p.) stimulated recombination of the *Pkd1* gene in mesenteric arteries of *Pkd1* ecKO mice but did not modify the *Pkd1* gene in arteries of *Pkd1^{fl/fl}* controls (*Figure 1—figure supplement 1*). Western blotting demonstrated that tamoxifen treatment

reduced PC-1 protein in mesenteric arteries of *Pkd1* ecKO mice to ~66.7% of that in *Pkd1*$^{fl/fl}$ control mice (**Figure 1A–B**). This reduction in PC-1 protein in *Pkd1* ecKO mice is expected given that arterial smooth muscle cells also express PC-1 (**Griffin et al., 1997**). In contrast, tamoxifen did not alter the expression of PC-2, small-conductance Ca$^{2+}$-activated K$^+$ (SK3) channels, intermediate-conductance Ca$^{2+}$-activated K$^+$ (IK) channels, TRP vanniloid 4 (TRPV4) channels, Piezo1 channels, endothelial NO synthase (eNOS), or G protein-coupled receptor 68 (GPR68) in arteries of *Pkd1* ecKO mice (**Figure 1A–B**). Immunofluorescence experiments demonstrated that PC-1 labeling was absent in ECs of *Pkd1* ecKO mouse mesenteric arteries imaged en face (**Figure 1C**). These data indicate that PC-1 expression is abolished in ECs of *Pkd1* ecKO mice.

## Flow stimulates a PC-1-dependent reduction in inward current in ECs

Patch-clamp electrophysiology was performed to investigate the regulation of plasma membrane currents by PC-1 in mesenteric artery ECs. Currents were recorded at steady-state voltage (–60 mV) using the whole-cell configuration with physiological ionic gradients. In a static bath, ECs of *Pkd1*$^{fl/fl}$ mice generated a mean steady-state inward current of ~–5.5 pA/pF (**Figure 1D and F**). Flow stimulated an initial transient increase in inward current of ~–1.3 pA/pF (**Figure 1D–E**). This transient increase in inward current was followed by a sustained reduction in inward current that reached steady state at ~–0.97 pA/pF in *Pkd1*$^{fl/fl}$ cells (**Figure 1D and F**). During flow, the removal of bath Ca$^{2+}$ increased mean inward current to ~–3.7 pA/pF in *Pkd1*$^{fl/fl}$ cells (**Figure 1D and F**). In a static bath, mean steady-state inward currents were similar in *Pkd1*$^{fl/fl}$ and *Pkd1* ecKO cells (**Figure 1D and F**). In contrast, the flow-activated transient inward current in *Pkd1* ecKO cells was ~15.4% of that in *Pkd1*$^{fl/fl}$ cells (**Figure 1D–E**). Similarly, the sustained flow-mediated reduction in inward current in *Pkd1* ecKO cells was ~42.2% of that in *Pkd1*$^{fl/fl}$ cells (**Figure 1D and F**). In the continuous presence of flow, the removal of bath Ca$^{2+}$ also resulted in a smaller increase in inward current in *Pkd1* ecKO cells than in *Pkd1*$^{fl/fl}$ cells (**Figure 1D and F**). Ca$^{2+}$ removal under flow increased inward current only ~–0.2 pA/pF in *Pkd1* ecKO cells, or ~7.1% of that in *Pkd1*$^{fl/fl}$ cells (**Figure 1D and F**). These data demonstrate that flow stimulates a PC-1-dependent biphasic response composed of an initial transient inward current followed by a sustained Ca$^{2+}$-dependent reduction in inward current in ECs.

## EC PC-1 contributes to flow-mediated arterial hyperpolarization

The flow-mediated, PC-1-dependent reduction in inward current in ECs suggested that PC-1 may regulate arterial potential, a major determinant of contractility (**Nelson et al., 1990**). Arterial potential was measured by impaling sharp glass microelectrodes into pressurized (80 mmHg) third-order mesenteric arteries of *Pkd1*$^{fl/fl}$ and *Pkd1* ecKO mice. In the absence of intraluminal flow, the membrane potentials of *Pkd1*$^{fl/fl}$ and *Pkd1* ecKO arteries were similar at either low (10 mmHg) or physiological (80 mmHg) pressures (**Figure 1G and H**). At 80 mmHg, intraluminal flow hyperpolarized *Pkd1*$^{fl/fl}$ arteries by ~10 mV, but *Pkd1* ecKO arteries by only ~2.6 mV, or ~27.1% of that in controls (**Figure 1G and H**). These data suggest that PC-1 expressed in ECs contributes to flow-mediated arterial hyperpolarization.

## PC-1 activates eNOS, IK channels, and SK channels in ECs to elicit vasodilation

The regulation of contractility by EC PC-1 was measured in pressurized (80 mmHg) third-order myogenic mesenteric arteries. Intravascular flow (15 dyn/cm$^2$) stimulated sustained and fully reversible dilations in mesenteric arteries of both *Pkd1*$^{fl/fl}$ and *Pkd1* ecKO mice (**Figure 2A**). In *Pkd1* ecKO arteries, dilations to on-off flow were ~34.8% of those in *Pkd1*$^{fl/fl}$ arteries (**Figure 2A and B**). In contrast, ACh-induced vasodilation was similar in *Pkd1*$^{fl/fl}$ and *Pkd1* ecKO arteries (**Figure 2B**; **Figure 2—figure supplement 1A**). To determine the functional flow range for PC-1 in ECs, we measured vasoregulation to shear stresses between 3 and 35 dyn/cm$^2$ (**Davies, 1995**). Cumulative stepwise increases in flow caused progressive dilation, with a maximum at 27 dyn/cm$^2$ in *Pkd1*$^{fl/fl}$ arteries (**Figure 2C and D**). Increasing shear stress above 27 dyn/cm$^2$ slightly attenuated maximal vasodilation (**Figure 2C and D**). The PC-1-sensitive component of flow-mediated dilation was calculated by subtracting responses in *Pkd1* ecKO arteries from those in *Pkd1*$^{fl/fl}$ arteries. Flow-mediated dilation in *Pkd1* ecKO arteries was attenuated over the entire shear-stress range to between ~38.0% and 50.0% of that in *Pkd1*$^{fl/fl}$ arteries (**Figure 2C and D**; **Figure 2—figure supplement 1B**). Myogenic tone, depolarization

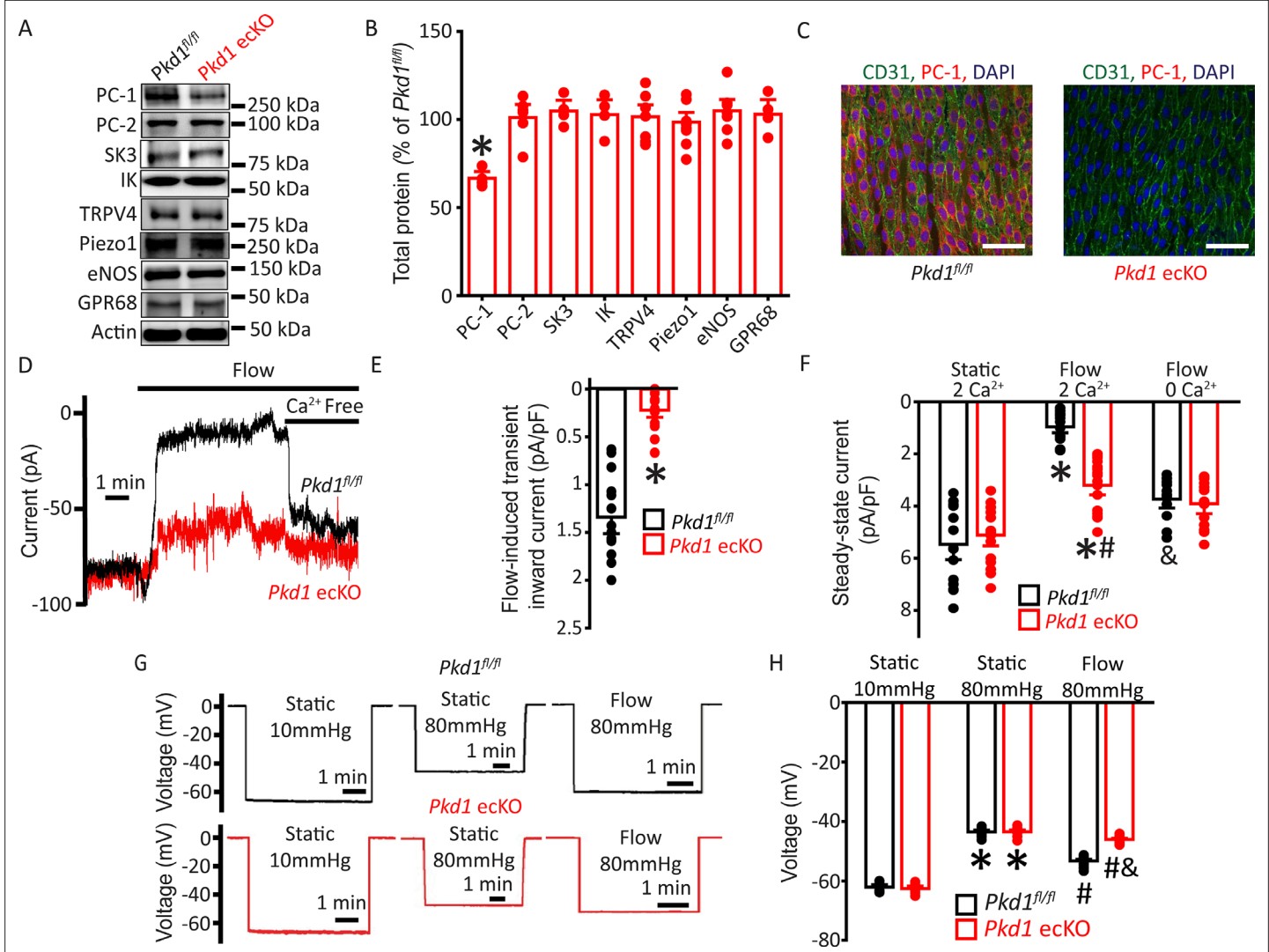

**Figure 1.** Flow stimulates PC-1-mediated, Ca²⁺-dependent currents in mesenteric artery endothelial cells (ECs) that elicit arterial hyperpolarization.
(**A**) Representative Western blots illustrating PC-1, PC-2, SK3, IK, TRPV4, Piezo1, eNOS, GPR68, and actin proteins in mesenteric arteries of *Pkd1ᶠˡ/ᶠˡ* and *Pkd1* ecKO mice. (**B**) Mean data for PC-1, PC-2, SK3, IK, TRPV4, Piezo1, eNOS, and GPR68, with n = 4, 6, 4, 4, 7, 8, 7, and 4, respectively. * indicates p<0.05. (**C**) En-face immunofluorescence illustrating that PC-1 (Alexa Fluor 546) is abolished in ECs of *Pkd1* ecKO mice mesenteric arteries (representative of eight arteries from *Pkd1ᶠˡ/ᶠˡ* and 7 *Pkd1* ecKO mice, respectively). CD31 (Alexa Fluor 488) and DAPI are also shown. Scale bars=50 µm. (**D**) Original recordings of steady-state current modulation by flow (10 ml/min) and effect of removing bath Ca²⁺ in ECs of *Pkd1ᶠˡ/ᶠˡ* and *Pkd1* ecKO mice voltage-clamped at −60 mV. (**E**) Mean data for flow-induced transient inward current density. n=15 for *Pkd1ᶠˡ/ᶠˡ* and n=16 for *Pkd1* ecKO. * indicates p<0.05 versus *Pkd1ᶠˡ/ᶠˡ*. (**F**) Mean data for steady-state current density (*Pkd1ᶠˡ/ᶠˡ*: static+Ca²⁺, n=15; flow+Ca²⁺, n=15; flow with Ca²⁺ free bath solution, n=12 and *Pkd1* ecKO: static+Ca²⁺, n=16; flow+Ca²⁺, n=16; flow with Ca²⁺ free bath, n=13). *p<0.05 versus static +2 mM Ca²⁺ in the same genotype, #p<0.05 versus *Pkd1ᶠˡ/ᶠˡ* under the same condition, &p<0.05 versus flow Ca²⁺ in the same genotype. (**G**) Original membrane potential recordings obtained using microelectrodes in pressurized (80 mmHg) mesenteric arteries of *Pkd1ᶠˡ/ᶠˡ* and *Pkd1* ecKO mice in static or flow (15 dyn/cm²) conditions. (**H**) Mean data (*Pkd1ᶠˡ/ᶠˡ*: 10 mmHg, n=8; 80 mmHg, n=14; 80 mmHg+ flow, n=19; *Pkd1* ecKO: 10 mmHg, n=9; 80 mmHg, n=14; 80 mmHg+ flow, n=18). *p<0.05 versus static at 10 mmHg in the same genotype. #p<0.05 for flow versus static at 80 mmHg in the same genotype. & indicates p<0.05 versus *Pkd1ᶠˡ/ᶠˡ* under the same condition.

The online version of this article includes the following figure supplement(s) for figure 1:

**Figure supplement 1.** Genomic PCR indicating that tamoxifen stimulated Cre-recombination in mesenteric arteries of *Pkd1ᶠˡ/ᶠˡ*: *Cdh5*(PAC)-CreERT2 mice.

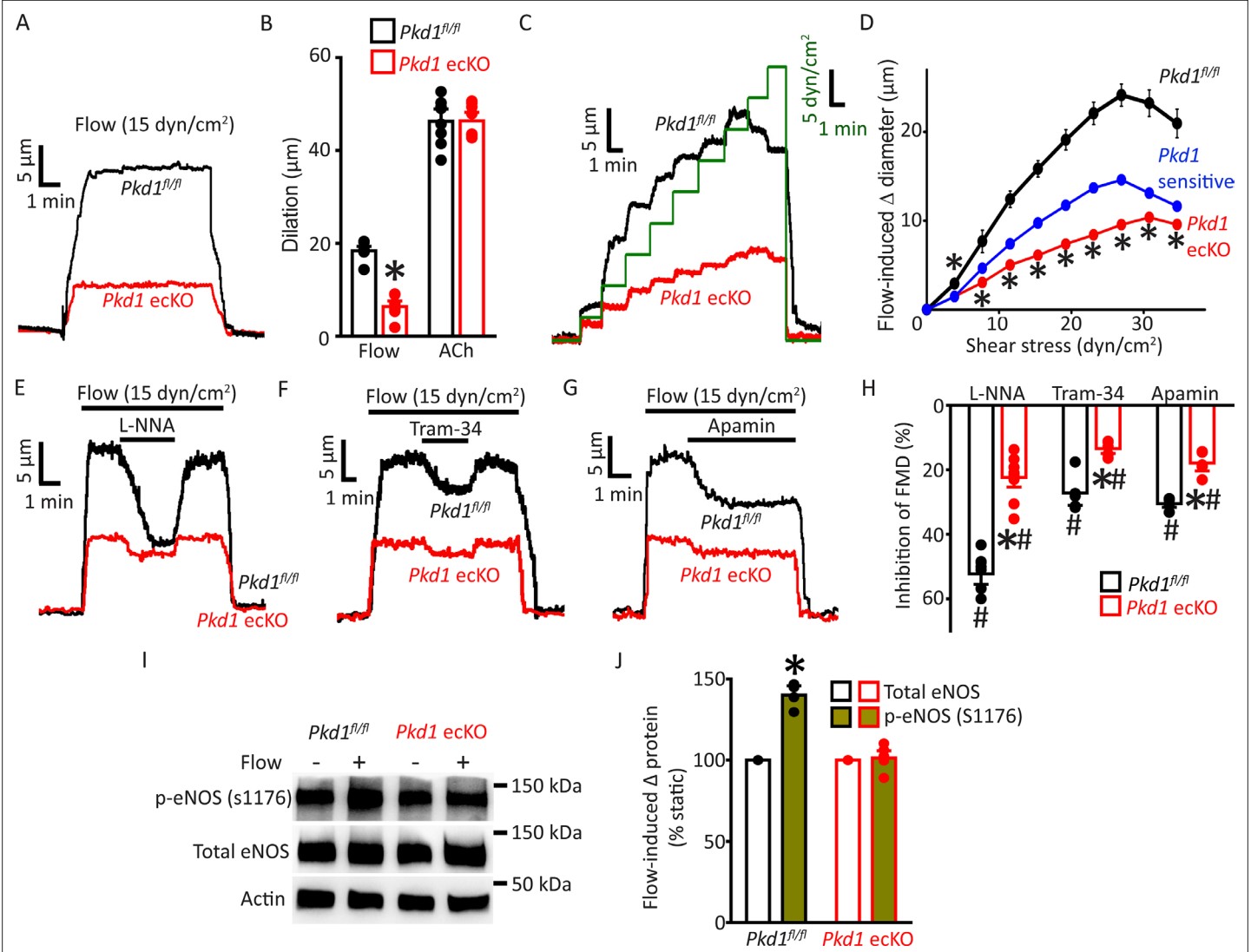

**Figure 2.** Endothelial cell PC-1 stimulates vasodilation via eNOS, IK channel, and SK channel activation. (**A**) Representative traces illustrating reversible flow-mediated dilation in pressurized (80 mmHg) mesenteric arteries of $Pkd1^{fl/fl}$ and $Pkd1$ ecKO mice. (**B**) Mean dilation to flow (15 dyn/cm$^2$) or ACh (10 μM). *p<0.05 versus $Pkd1^{fl/fl}$. n=8 for each data set. (**C**) Representative diameter changes to stepwise increases in intravascular flow in pressurized (80 mmHg) mesenteric arteries from $Pkd1^{fl/fl}$ and $Pkd1$ ecKO mice. (**D**) Mean data. The $Pkd1$-sensitive component of flow-mediated vasodilation is shown in blue. n=4 each for $Pkd1^{fl/fl}$ and $Pkd1$ ecKO. *p<0.05 versus $Pkd1^{fl/fl}$. (**E–G**) Regulation of flow (15 dyn/cm$^2$)-mediated dilation by L-NNA (10 μM), apamin (300 nM), and Tram-34 (300 nM) in pressurized (80 mmHg) mesenteric arteries of $Pkd1^{fl/fl}$ and $Pkd1$ ecKO mice. (**H**) Mean data for inhibition of flow-mediated vasodilation (FMD) by L-NNA ($Pkd1^{fl/fl}$ n=8, $Pkd1$ ecKO n=10), Tram-34 ($Pkd1^{fl/fl}$ n=5, $Pkd1$ ecKO n=5), and apamin ($Pkd1^{fl/fl}$ n=5, $Pkd1$ ecKO n=5). Symbols illustrate #p<0.05 versus flow in the same genotype and *p<0.05 versus $Pkd1^{fl/fl}$ in the same condition. (**I**) Original Western blots illustrating effects of flow (15 dyn/cm$^2$, 5 min, 37°C) on p-eNOS (S1176) and total eNOS proteins in $Pkd1^{fl/fl}$ and $Pkd1$ ecKO mesenteric arteries. (**J**) Mean data for flow-induced change (Δ) in proteins. $Pkd1^{fl/fl}$ n=4, $Pkd1$ ecKO n=6. * indicates p<0.05 versus static in the same genotype.

The online version of this article includes the following figure supplement(s) for figure 2:

**Figure supplement 1.** Depolarization-induced vasoconstriction and passive diameter are similar in $Pkd1^{fl/fl}$ and $Pkd1$ ecKO arteries.

(60 mM K$^+$)-induced constriction and passive diameter were similar in $Pkd1^{fl/fl}$ and $Pkd1$ ecKO arteries (**Figure 2—figure supplement 1C-F**). These data indicate that a broad intravascular flow range activates PC-1 in ECs to induce vasodilation.

Ca$^{2+}$ influx activates eNOS, IK channels, and SK channels in ECs, leading to vasodilation (**Vane, 1993**; **Garland et al., 2011**). Next, we studied the contributions of each of these proteins to flow-mediated, PC-1-dependent vasodilation. L-NNA, a NOS inhibitor, Tram-34, an IK channel blocker, and apamin, a SK channel inhibitor, inhibited flow-mediated dilation in pressurized (80 mmHg) myogenic $Pkd1^{fl/fl}$

arteries (*Figure 2E–H*). PC-1 knockout reduced the L-NNA-, Tram-34-, and apamin-sensitive components of flow-mediated dilation to ~21.6%, 13.5%, and 18.6% of those in *Pkd1^{fl/fl}* arteries, respectively (*Figure 2E–H*). We have previously shown that in the absence of intravascular flow, L-NNA, Tram-34, and apamin alone do not alter the diameter of pressurized mesenteric arteries (*MacKay et al., 2020*). These data indicate that flow stimulates PC-1-mediated dilation via NOS, IK channel, and SK channel activation in ECs.

Flow activates eNOS in ECs, but the involvement of PC-1 in mediating this response is unclear (*Fleming, 2010*; *Balligand et al., 2009*; *Garcia and Sessa, 2019*). Phosphorylation of eNOS at serine 1176 (p-eNOS (S1176)) leads to its activation (*Fulton et al., 1999*; *Dimmeler et al., 1999*). Western blotting experiments indicated that intravascular flow increased p-eNOS (S1176) protein ~1.40-fold in *Pkd1^{fl/fl}* arteries, but did not alter p-eNOS in *Pkd1* ecKO arteries (*Figure 2I and J*). In contrast, flow did not alter total eNOS in either genotype (*Figure 2I and J*). These data indicate that flow stimulates PC-1 in ECs, leading to eNOS, IK channel, and SK channel activation, which produces vasodilation.

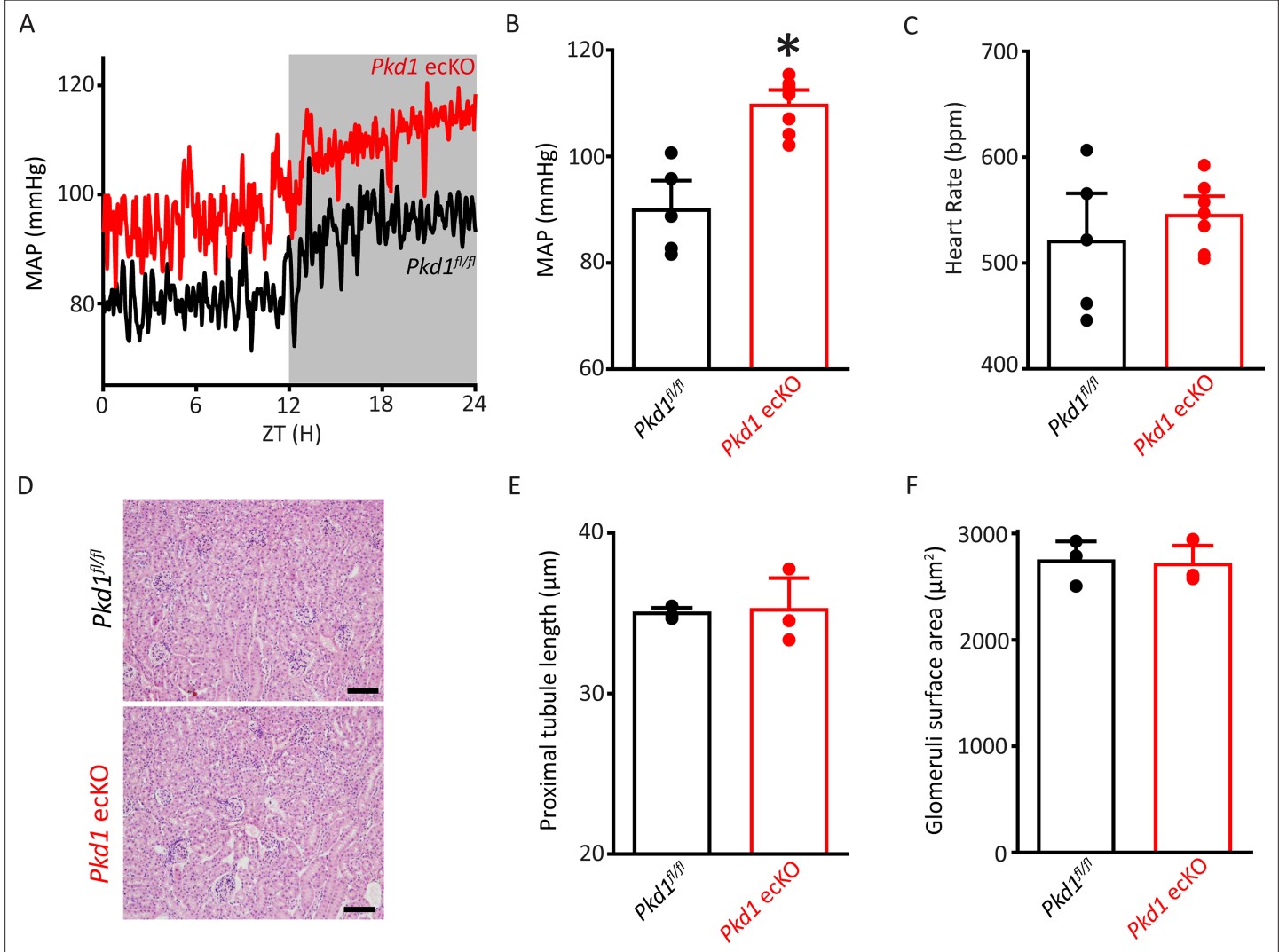

**Figure 3.** *Pkd1* ecKO mice are hypertensive with normal kidney anatomy. (**A**) Blood pressure recordings obtained over 24 hr in a *Pkd1^{fl/fl}* and *Pkd1* ecKO mouse. (**B**) Mean arterial pressures (MAPs) in *Pkd1^{fl/fl}* (n=5) and *Pkd1* ecKO (n=7) mice. *p<0.05 versus *Pkd1^{fl/fl}*. (**C**) Mean heart rate (HR). *Pkd1^{fl/fl}*, n=5, *Pkd1* ecKO, n=7. (**D**) Images of H&E-stained kidney cortex used for histological measurements. Scale bars=100 µm (**E**) Mean proximal tubule length. n=15 proximal tubules measured in each mouse, n=3 mice. (**F**) Mean glomeruli surface area. n=75 glomeruli measured from each mouse, n=3 mice. H&E, hematoxylin and eosin.

The online version of this article includes the following figure supplement(s) for figure 3:

**Figure supplement 1.** Diastolic and systolic blood pressures are higher in *Pkd1* ecKO mice.

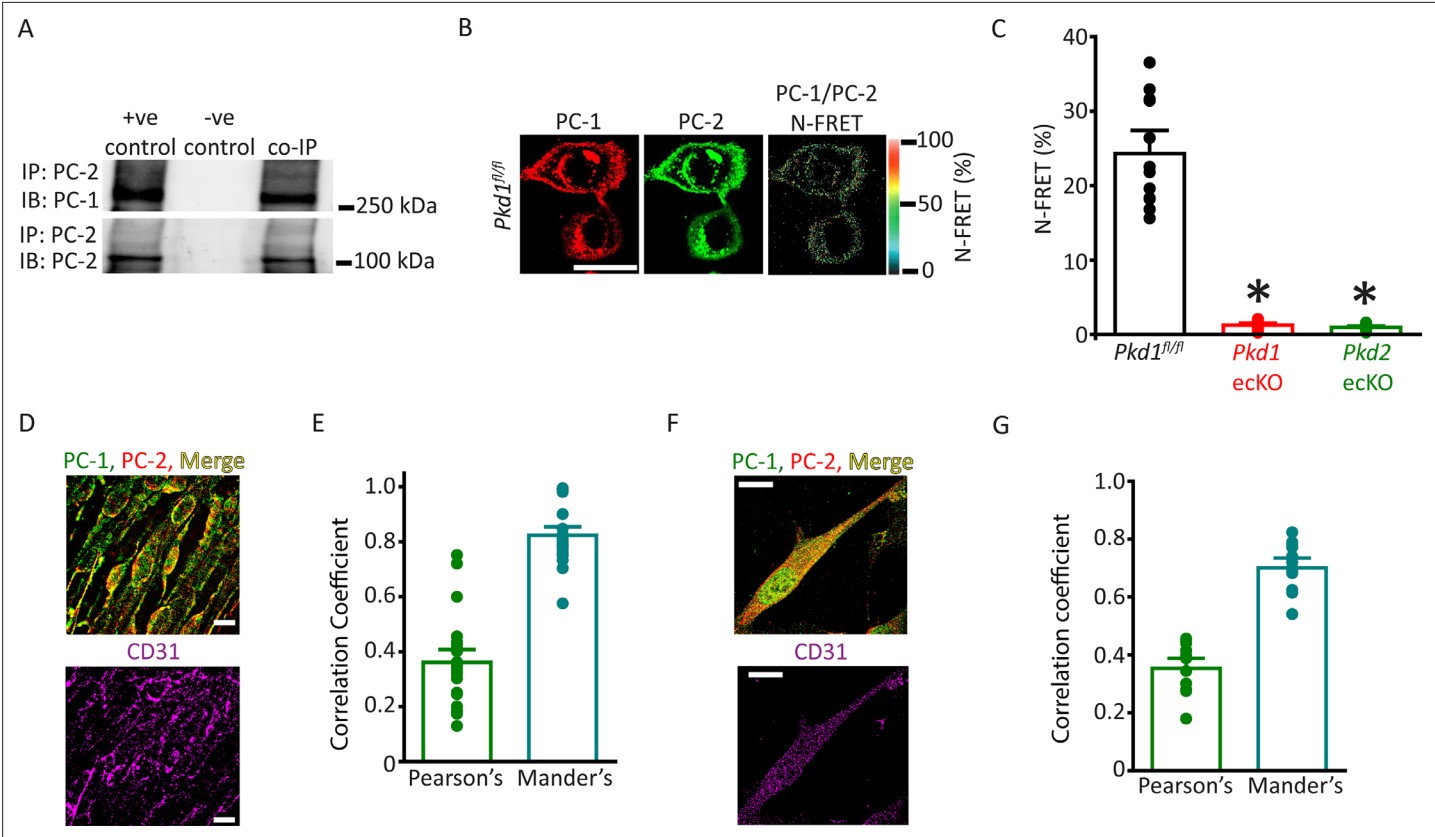

**Figure 4.** PC-1 and PC-2 coassemble and colocalize in endothelial cells (ECs). (**A**) Representative Western blots illustrating the immunoblot (IB) detection of both PC-1 and PC-2 in PC-2 immunoprecipitate (IP) (n=5). (**B**) PC-1 (Alexa546) and PC-2 (Alexa488) antibody labeling generate FRET in mesenteric artery ECs of *Pkd1fl/fl* mice that are abolished in ECs of *Pkd1* ecKO and *Pkd2* ecKO mice. Scale bar=10 µm. (**C**) Mean data for immunoFRET experiments (*Pkd1fl/fl* n=12, *Pkd1* ecKO n=10, *Pkd2* ecKO n=10). *p<0.05 versus *Pkd1fl/fl*. (**D**) Lattice SIM images of PC-1, PC-2, and CD31 immunofluorescence in the same ECs of an en face mesenteric artery. The merged image is also shown with yellow pixels illustrating colocalization of PC-1 and PC-2. Scale bars=10 µm. (**E**) Mean data for PC-1 and PC-2 colocalization using both Pearson's and Mander's correlation coefficients. n=25 images, 12 arteries and 6 mice for each data set. (**F**) Lattice SIM image of PC-1 and PC-2 immunofluorescence in a mesenteric artery EC. Yellow pixels illustrate PC-1 to PC-2 colocalization. Scale bar=10 µm. (**G**) Mean data for PC-1 and PC-2 colocalization when using Pearson's and Mander's correlation coefficients. n=13, four mice for each data set. FRET, fluorescence energy transfer.

The online version of this article includes the following figure supplement(s) for figure 4:

**Figure supplement 1.** PC-1 or PC-2 knockout abolishes immunoFRET in endothelial cells (ECs).

## EC PC-1 reduces systemic blood pressure

Radiotelemetry measurements were performed to measure blood pressure in freely moving, conscious mice. Mean arterial pressure (MAP) was ~89.9 mmHg in *Pkd1fl/fl* mice and ~109.6 mmHg in *Pkd1* ecKO mice, or 21.9% higher (*Figure 3A–B*). Underlying this increase in MAP were higher systolic and diastolic blood pressures in *Pkd1* ecKO mice (*Figure 3—figure supplement 1A*). In contrast, heart rate and activity were similar in *Pkd1fl/fl* and *Pkd1* ecKO mice, as were proximal tubule diameter and glomerular surface area measured in hematoxylin and eosin (H&E)-stained kidney sections (*Figure 3C–F*; *Figure 3—figure supplement 1B*). These results suggest that flow stimulates PC-1 in ECs, leading to vasodilation and a reduction in systemic blood pressure.

## PC-1 and PC-2 coassemble and colocalize in ECs

The vascular phenotype we describe here for *Pkd1* ecKO mice is similar to that of *Pkd2* ecKO mice (*MacKay et al., 2020*). Thus, we tested the hypothesis that PC-1 and PC-2 are components of the same flow-sensitive signaling pathway in ECs. PC-1 and PC-2 coimmunoprecipitated in mesenteric artery lysate, suggesting they coassemble (*Figure 4A*). Next, we used several different imaging techniques to measure the spatial proximity of PC-1 and PC-2 proteins in ECs. Immunofluorescence energy

transfer (immunoFRET) microscopy using Alexa Fluor 546 and Alexa Fluor 488-tagged secondary antibodies bound to PC-1 and PC-2 primary antibodies, respectively, generated mean N-FRET of ~24.3% in ECs of *Pkd1^{fl/fl}* mice (*Figure 4B and C*). In contrast, N-FRET was only ~1.3% and 0.9% in *Pkd1* ecKO and *Pkd2* ecKO ECs, respectively, when using the same labeling procedure (*Figure 4C*; *Figure 4— figure supplement 1A, B*). Given that the Förster coefficient of the Alexa Fluor pair used is ~6.3 nm, these data suggest PC-1 and PC-2 locate in close spatial proximity. Lattice structured illumination super-resolution microscopy (Lattice-SIM) was used to measure colocalization of PC-1 and PC-2 in ECs of en face mesenteric arteries and in isolated mesenteric artery endothelial cells (*Figure 4D–G*). ECs were identified through immunolabeling of CD31, an EC-specific marker (*Figure 4D and F*). Analysis of these data using both Pearson's and Mander's coefficients also indicated that PC-1 and PC-2 colocalize in ECs (*Figure 4E and G*; *Bolte and Cordelières, 2006*).

## PC-1 and PC-2 surface clusters exhibit nanoscale colocalization and interdependency in ECs

Single-molecule localization microscopy (SMLM) in combination with total internal reflectance (TIRF) imaging (SMLM-TIRF) was used to measure the properties and nanoscale proximity of PC-1 and PC-2 protein clusters in the plasma membrane of ECs. Imaging was performed on cells which labeled for CD31, an endothelial-specific marker. Localization precision of the Alexa Fluor 488 and Alexa Fluor 647 fluorophores on secondary antibodies that were used for SMLM-TIRF were 29.6±0.6 (n=53) and 24.6±0.7 (n=45) nm, respectively, when imaged in ECs (*Figure 5—figure supplement 1A, B*). Discrete clusters of PC-1 and PC-2 were observed in the plasma membrane of *Pkd1^{fl/fl}* ECs (*Figure 5A*). PC-1 and PC-2 clusters exhibited similar densities (*Figure 5B*). The sizes (areas) of individual PC-1 and PC-2 clusters were exponentially distributed, with means of ~3702.5 and 2157.1 nm², respectively (*Figure 5C*; *Figure 5—figure supplement 2A, B*). PC-2 clusters were smaller than PC-1 clusters (*Figure 5C*; *Figure 5—figure supplement 2A*). Histograms were constructed that contained the distance between the center of each PC-1 cluster and that of its nearest PC-2 neighbor. These data were exponentially distributed (*Figure 5—figure supplement 2C*). The mean PC-1 to PC-2 intercentroid distance was ~126.8 nm in *Pkd1^{fl/fl}* cells (*Figure 5D*). Approximately 27.0% of PC-1 clusters overlapped with a PC-2 cluster and ~26.6% of PC-2 clusters overlapped with a PC-1 cluster in *Pkd1^{fl/fl}* ECs (*Figure 5E and F*; *Figure 5—figure supplement 2D*). When experimental data were randomized using Costes' simulation algorithm (*Bolte and Cordelières, 2006*), PC-1 to PC-2 and PC-2 to PC-1 overlap were only ~7.8% and 7.2%, respectively, in *Pkd1^{fl/fl}* cells (*Figure 5E and F*; *Figure 5—figure supplement 2D*). Flow did not alter PC-1 or PC-2 cluster density, PC-1 or PC-2 cluster size, the distance between PC-1 and PC-2 centroids, or their overlap (*Figure 5B–F*; *Figure 5—figure supplement 2D, E*). In *Pkd1* ecKO cells, the densities of PC-1 and PC-2 clusters were far lower, at ~15.7% and 18.3%, respectively, of those in ECs of *Pkd1^{fl/fl}* mice (*Figure 5A and B*). The mean sizes of PC-1 and PC-2 clusters in *Pkd1* ecKO ECs were ~17.7% and 28.3% of those, respectively, in *Pkd1^{fl/fl}* cells (*Figure 5C*; *Figure 5—figure supplement 2A, B*). The mean PC-1 to PC-2 distance was ~5.79-fold larger in *Pkd1* ecKO than *Pkd1^{fl/fl}* cells (*Figure 5D*; *Figure 5—figure supplement 2C*). Furthermore, only ~1.2% of PC-1 clusters overlapped with a PC-2 cluster in *Pkd1* ecKO cells, with similar results for PC-2 to PC-1 overlap (*Figure 5E and F*; *Figure 5—figure supplement 2D*). Costes' randomization of data did not alter PC-1 to PC-2 or PC-2 to PC-1 overlap in *Pkd1* ecKO cells. These data indicate that: (1) PC-1 and PC-2 surface protein clusters are colocalized in ECs, (2) PC-1 or PC-2 knockout inhibits surface expression of both PC-1 and PC-2 proteins, and (3) fluorescent clusters in *Pkd1* ecKO cells are likely due to nonspecific labeling by secondary antibodies.

To determine whether data obtained using SMLM were specific to ECs of *Pkd1^{fl/fl}* and *Pkd1* ecKO mice, we performed similar experiments using tamoxifen-inducible, EC-specific PC-2 knockout (*Pkd2* ecKO) mice, and their controls (*Pkd2^{fl/fl}*). The mean densities and sizes of PC-1 and PC-2 clusters, PC-2 to PC-1 intercluster distance, PC-1 to PC-2 overlap, and PC-2 to PC-1 overlap were all similar in ECs of *Pkd2^{fl/fl}* mice and *Pkd1^{fl/fl}* mice (*Figure 5—figure supplement 3A-H* and *Figure 5—figure supplement 4A, B*). Similarly to observations in *Pkd1^{fl/fl}* cells, flow did not alter the densities, sizes, intercluster distances, or overlap of PC-1 or PC-2 clusters in *Pkd2^{fl/fl}* cells (*Figure 5—figure supplement 3B, E, G, H* and *Figure 5—figure supplement 4A, B*). In *Pkd2* ecKO cells, PC-1 and PC-2 cluster densities were far lower, at ~12.7% and 11.1% of those in *Pkd2^{fl/fl}* cells (*Figure 5—figure supplement 3A, B*). PC-1 and PC-2 clusters were smaller and the mean distance from PC-2 clusters to their nearest PC-1

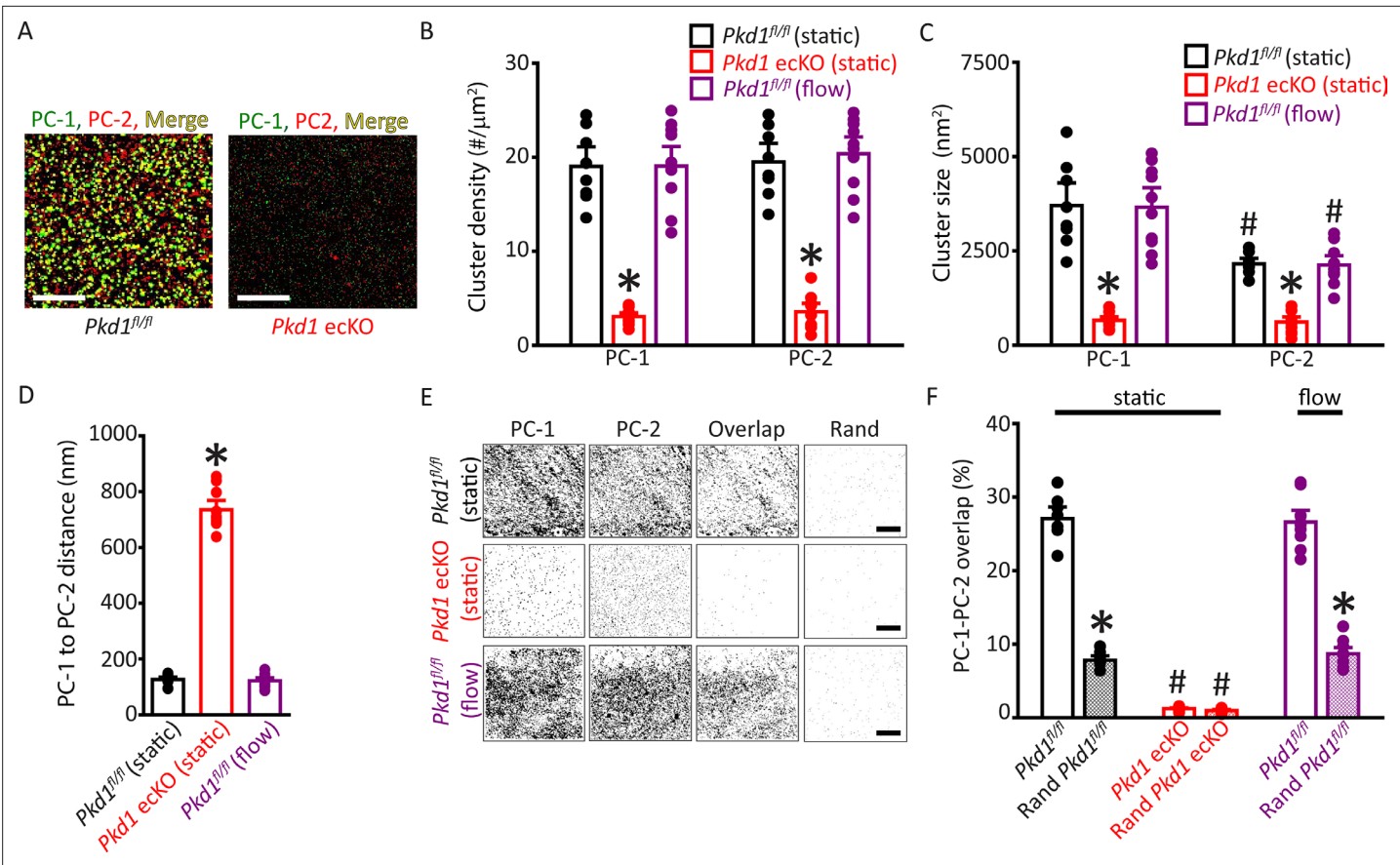

**Figure 5.** Plasma membrane PC-1 and PC-2 clusters colocalize in endothelial cells (ECs). (**A**) TIRF-SMLM images of PC-1 and PC-2 surface clusters in a *Pkd1^fl/fl^* and *Pkd1* ecKO EC. Scale bars=5 µm. (**B**) Mean data for PC-1 and PC-2 cluster density measured in *Pkd1^fl/fl^* and *Pkd1* ecKO ECs under static and flow (10 ml/min) conditions. n=8 for *Pkd1^fl/fl^* (static), n=10 for *Pkd1* ecKO (static), and n=10 for *Pkd1^fl/fl^* (flow). *p<0.05 versus *Pkd1^fl/fl^* (static). (**C**) Mean data for PC-1 and PC-2 cluster sizes measured in *Pkd1^fl/fl^* and *Pkd1* ecKO ECs under static and flow (10 ml/min) conditions. n=8 for *Pkd1^fl/fl^* (static), n=10 for *Pkd1* ecKO (static), and n=10 for *Pkd1^fl/fl^* (flow). *p<0.05 versus respective floxed control in static, #p<0.05 versus PC-1 cluster size in the same genotype under the same condition. (**D**) Mean data for PC-1 to PC-2 nearest-neighbor analysis. n=8 for *Pkd1^fl/fl^* (static), n=10 for *Pkd1* ecKO (static) and n=10 for *Pkd1^fl/fl^* (flow). *p<0.05 versus *Pkd1^fl/fl^* static and *Pkd1^fl/fl^* flow. (**E**) TIRF-SMLM images of PC-1 and PC-2 clusters, overlap of PC-1 and PC-2 data and overlap of PC-1 and PC-2 data following Coste's randomization (Rand) simulation in *Pkd1^fl/fl^* and *Pkd1* ecKO cells. Scale bars=5 µm. (**F**) Mean experimental and Costes' randomized (Rand) data for PC-1 to PC-2 overlap in *Pkd1^fl/fl^* cells in static and flow and *Pkd1* ecKO cells in static. n=8 for *Pkd1^fl/fl^* (static), n=10 for *Pkd1* ecKO (static), and n=10 for *Pkd1^fl/fl^* (flow). *p<0.05 versus respective floxed control in static condition, #p<0.05 versus *Pkd1^fl/fl^* static. SMLM, single-molecule localization microscopy; TIRF, total internal reflection fluorescence.

The online version of this article includes the following figure supplement(s) for figure 5:

**Figure supplement 1.** Localization precision of fluorophores used in SMLM experiments.

**Figure supplement 2.** Properties of PC-1 and PC-2 clusters and their spatial proximity and overlap in *Pkd1^fl/fl^* and *Pkd1* ecKO endothelial cells (ECs).

**Figure supplement 3.** Properties of PC-1 and PC-2 surface clusters in endothelial cells (ECs) of *Pkd2^fl/fl^* and *Pkd2* ecKO mice.

**Figure supplement 4.** Analysis of PC-1 and PC-2 cluster overlap in endothelial cells of *Pkd2^fl/fl^* and *Pkd2* ecKO mice.

neighbor was far greater in *Pkd2* ecKO cells than in *Pkd2^fl/fl^* cells (***Figure 5—figure supplement 3C-H***). PC-1 to PC-2 and PC-2 to PC-1 overlap were also lower in *Pkd2* ecKO than *Pkd2^fl/fl^* cells (***Figure 5— figure supplement 4A, B***). These data demonstrate that PC-1 and PC-2 surface clusters colocalize in ECs. Knockout of either PC-1 or PC-2 inhibits the surface expression of both PC-1 and PC-2 proteins, supporting their interdependency. Fluorescent clusters observed in *Pkd1* ecKO and *Pkd2* ecKO cells appear to represent nonspecific secondary antibody labeling.

## Flow-mediated vasodilation is similarly attenuated in arteries of *Pkd1* ecKO and *Pkd1/Pkd2* ecKO mice

Next, we investigated the functional interdependency of PC-1 and PC-2 in flow-mediated vasodilation. *Pkd1*$^{fl/fl}$:*Cdh5(PAC)-CreERT2* and *Pkd2*$^{fl/fl}$:*Cdh5(PAC)-CreERT2* mice were crossed to generate a *Pkd1*$^{fl/fl}$/*Pkd2*$^{fl/fl}$:*Cdh5(PAC)-CreERT2* line. Control Cre-negative *Pkd1*$^{fl/fl}$/*Pkd2*$^{fl/fl}$ mice were generated using a similar breeding strategy. Tamoxifen-treatment reduced PC-1 and PC-2 proteins in mesenteric arteries of *Pkd1*$^{fl/fl}$/*Pkd2*$^{fl/fl}$:*Cdh5(PAC)-CreERT2* mice to ~60.9% and 66.1%, respectively, of those in *Pkd1*$^{fl/fl}$/*Pkd2*$^{fl/fl}$ mice (*Figure 6A and B*, p<0.05). In contrast, eNOS was similar in *Pkd1/Pkd2* ecKO arteries and *Pkd1*$^{fl/fl}$/*Pkd2*$^{fl/fl}$ arteries (*Figure 6A and B*). Intravascular flow stimulated vasodilation in pressurized *Pkd1*$^{fl/fl}$/*Pkd2*$^{fl/fl}$ arteries that was similar in magnitude to that in arteries of both *Pkd1*$^{fl/fl}$ and *Pkd2*$^{fl/fl}$ mice (*Figure 2A–C*; *Figure 6C and D* and *MacKay et al., 2020*). In *Pkd1/Pkd2* ecKO mouse arteries, mean flow-mediated vasodilation at 15 and 23 dyn/cm$^2$ were ~38.8% and 50.4% of those in *Pkd1*$^{fl/fl}$/*Pkd2*$^{fl/fl}$ arteries, respectively (*Figure 6C and D*). This inhibition of flow-mediated dilation is similar to that in arteries of *Pkd1* ecKO and *Pkd2* ecKO mice when compared to their respective controls (*Figures 2A–C, 6C and D* and *MacKay et al., 2020*). In contrast, 60 mM K$^+$-induced constriction, myogenic tone, dilations to ACh, or sodium nitroprusside (SNP), a NO donor, and passive diameter were similar in *Pkd1*$^{fl/fl}$/*Pkd2*$^{fl/fl}$ and *Pkd1/Pkd2* ecKO arteries (*Figure 6D*; *Figure 6—figure supplement 1A-E*). Collectively, these data indicate that intravascular flow stimulates vasodilation via a mechanism that involves both PC-1 and PC-2 in ECs.

## PC-1 knockout, PC-2 knockout, and coiled-coil domain peptides similarly inhibit flow-activated nonselective cation current (I$_{Cat}$) in endothelial cells

Recombinant PC-1 and PC-2 can physically interact via several domains, including their C-terminal coiled-coils (*Qian et al., 1997*; *Zhu et al., 2011*; *Yu et al., 2009*; *Tsiokas et al., 1997*). Next, we investigated the functional significance of the coiled-coil domains present in PC-1 and PC-2 to flow-mediated signaling. As PC-2 is a nonselective cation channel, we performed these experiments by isolating and measuring I$_{Cat}$ in ECs. I$_{Cat}$ was measured using solutions that inhibit K$^+$ channels and a bath solution that was Ca$^{2+}$-free to prevent Ca$^{2+}$ influx-dependent activation of channels. At a steady-state holding potential of –60 mV, flow reversibly stimulated sustained inward I$_{Cat}$s that were of similar amplitude in *Pkd1*$^{fl/fl}$ and *Pkd2*$^{fl/fl}$ ECs (*Figure 6E–G*). In *Pkd1* ecKO ECs, mean flow-activated I$_{Cat}$ was ~23.3% of that in *Pkd1*$^{fl/fl}$ controls (*Figure 6E and G*). Flow-activated I$_{Cat}$ in *Pkd2* ecKO cells was similarly smaller, at ~22.5% of that in *Pkd2*$^{fl/fl}$ cells (*Figure 6E and G*). These data indicate that PC-1 and PC-2 similarly contribute to flow-activated I$_{Cat}$ in ECs. These data also suggest that when using physiological ionic gradients, flow activates PC-1/PC-2 to induce a transient inward current which then stimulates K$^+$ channels to produce the sustained reduction in inward current (*Figure 1D–F* and *MacKay et al., 2020*).

Peptides were constructed that correspond to regions within the C-terminal coiled-coil domains in PC-1 (amino acids 4216–4233) and PC-2 (amino acids 874–883) that physically interact (*Qian et al., 1997*; *Zhu et al., 2011*; *Yu et al., 2009*; *Hanaoka et al., 2000*). Intracellular introduction via the pipette solution of either the PC-1 or PC-2 coiled-coil domain peptide similarly reduced flow-activated I$_{Cat}$s to between ~30.1% and 29% of control currents in both *Pkd1*$^{fl/fl}$ and *Pkd2*$^{fl/fl}$ ECs (*Figure 6F and G*). In contrast, scrambled PC-1 and PC-2 peptides did not alter flow-activated I$_{Cat}$s (*Figure 6G*; *Figure 6—figure supplement 2*). Lowering bath Na$^+$ concentration inhibited flow-activated I$_{Cat}$ in all genotypes and under all conditions. These data indicate that PC-1 and PC-2 are interdependent for flow to activate I$_{Cat}$ in ECs. Data also suggest that physical coupling of PC-1 and PC-2 C-termini is necessary for flow to activate I$_{Cat}$.

## Discussion

Here, we investigated the regulation of arterial contractility by PC-1 and the potential involvement of PC-2 in mediating responses in ECs. Our data show that flow stimulates PC-1-dependent cation currents in ECs that induce arterial hyperpolarization, vasodilation, and a reduction in blood pressure. PC-1-dependent vasodilation occurs over the entire functional shear stress range due to eNOS, IK channel, and SK channel activation. PC-1 and PC-2 proteins coassemble and their surface clusters

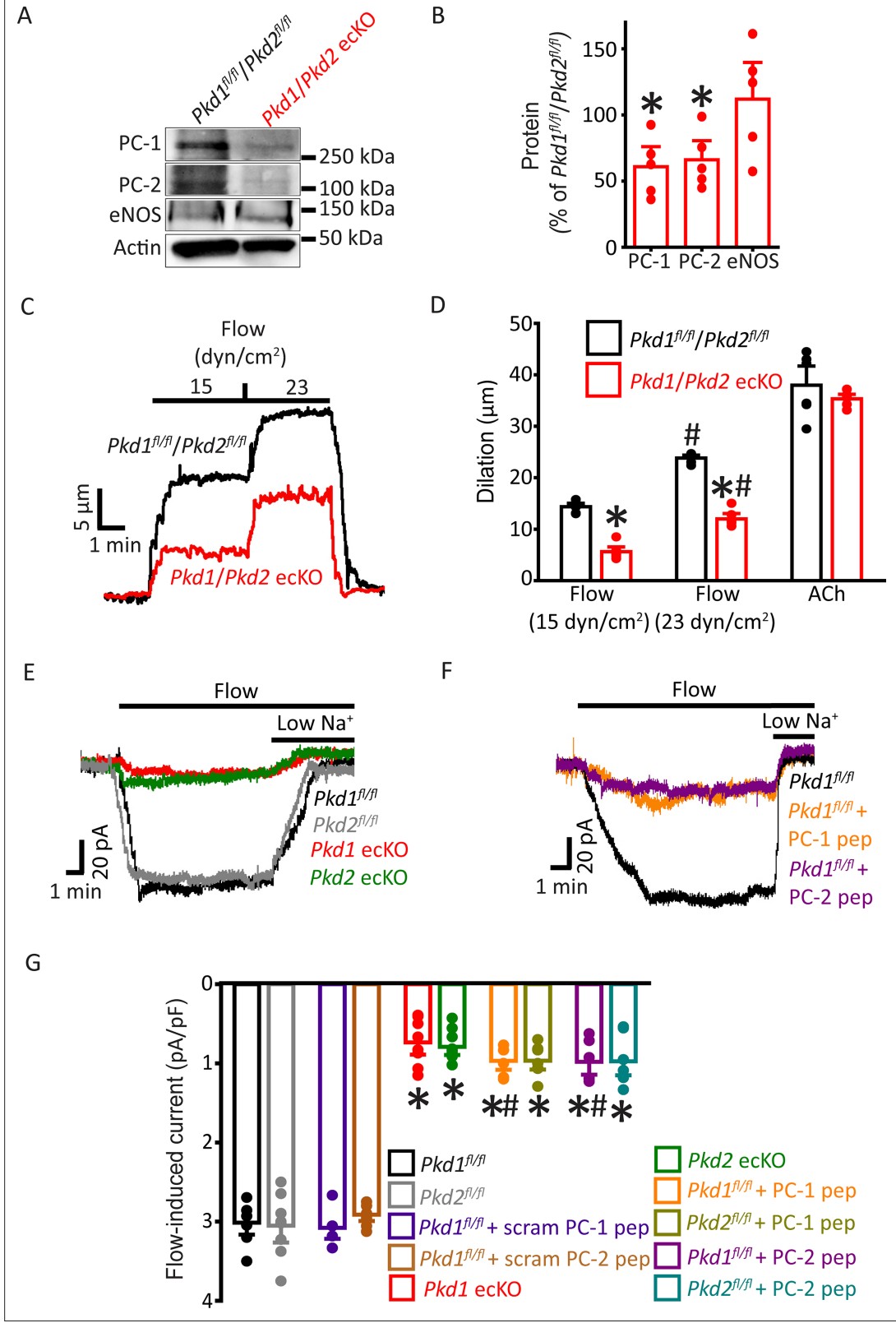

**Figure 6.** PC-1 and PC-2 are interdependent for flow-mediated vasodilation and $I_{Cat}$ activation in mesenteric artery endothelial cells (ECs).
(**A**) Representative Western blots of PC-1, PC-2, and eNOS in mesenteric arteries of $Pkd1^{fl/fl}/Pkd2^{fl/fl}$ and $Pkd1/Pkd2$ ecKO mice. (**B**) Mean data. *p<0.05 versus $Pkd1^{fl/fl}/Pkd2^{fl/fl}$. n=5 for each data set. (**C**) Representative traces illustrating flow-mediated dilation by 15 and 23 dyn/cm² shear stress in pressurized (80 mmHg) mesenteric arteries of $Pkd1^{fl/fl}/Pkd2^{fl/fl}$ and $Pkd1/Pkd2$ ecKO mice. (**D**) Mean dilation to flow (15 and 23 dyn/cm²) or ACh

*Figure 6 continued on next page*

*Figure 6 continued*

(10 µM). n=6 each for 15, 23 dyn/cm$^2$, and ACh. *p<0.05 versus *Pkd1$^{fl/fl}$/Pkd2$^{fl/fl}$*. #p<0.05 versus 15 dyn/cm$^2$ in the same genotype. (**E**) Original recordings illustrating that flow (10 ml/min) activates I$_{Cat}$s in *Pkd1$^{fl/fl}$* and *Pkd2$^{fl/fl}$* ECs that are similarly attenuated in PC-1 ecKO and PC-2 ecKO ECs. (**F**) Intracellular introduction via the patch pipette of either a PC-1 or PC-2 C-terminus coiled-coil domain peptide reduces flow-induced I$_{Cat}$s in *Pkd1$^{fl/fl}$* ECs. Traces shown in (**E**) and (**F**) are recorded from different ECs. (**G**) Mean data. PC-1 pep and PC-2 pep indicate peptides corresponding to the coiled-coil domains in PC-1 and PC-2, respectively. *Pkd1$^{fl/fl}$* n=7, *Pkd2$^{fl/fl}$* n=8, *Pkd1$^{fl/fl}$*+ scrambled (scram) PC-1 pep n=6, *Pkd1$^{fl/fl}$*+ scram PC-2 pep n=7, *Pkd1* ecKO n=8, *Pkd2* ecKO n=9, *Pkd1$^{fl/fl}$*+ PC-1 peptide n=6, *Pkd2$^{fl/fl}$*+ PC-1 peptide n=7, *Pkd1$^{fl/fl}$*+ PC-2 peptide n=6, and *Pkd2$^{fl/fl}$*+ PC-2 peptide n=7. *p<0.05 versus no peptide in same genotype. #p<0.05 versus respective scrambled peptide in same genotype.

The online version of this article includes the following figure supplement(s) for figure 6:

**Figure supplement 1.** Smooth muscle-specific vasoconstriction and vasodilation and passive diameter are unaltered in *Pkd1*/*Pkd2* ecKO mouse arteries.

**Figure supplement 2.** Scrambled PC-1 and PC-2 peptides do not alter flow-activated I$_{Cat}$ in endothelial cells (ECs).

colocalize. Knockout of either PC-1 or PC-2 abolishes surface expression of both proteins. Flow activates a PC-1- and PC-2-dependent I$_{Cat}$ that is inhibited by peptides corresponding to the C-terminus coiled-coil domains on either polycystin protein. These data demonstrate that PC-1 regulates arterial contractility by forming an interdependent complex with PC-2 in ECs. Flow stimulates plasma membrane-localized PC-1/PC-2 clusters, leading to eNOS, IK channel, and SK channel activation, vasodilation, and a reduction in blood pressure.

Intraluminal flow stimulates ECs to induce vasodilation, but signaling mechanisms involved are poorly understood. Data here demonstrate that flow stimulates signaling mechanisms in ECs that are similarly dependent on PC-1 and PC-2. When combined with the results of our earlier study, flow-mediated signaling and vasodilation are similarly attenuated in ECs and arteries of *Pkd1* ecKO, *Pkd2* ecKO, and *Pkd1*/*Pkd2* ecKO mice, further strengthening this conclusion (**MacKay et al., 2020**). Specific results include that inducible, EC-specific knockout of either PC-1 or PC-2 similarly inhibits flow-mediated: (1) biphasic currents in ECs, (2) Ca$^{2+}$ influx-dependent cation currents in ECs, (3) arterial hyperpolarization, (4) vasodilation over a broad shear stress range, and (5) vasodilation that occurs via eNOS, IK channel, and SK channel activation. We also show that PC-1 knockout elevates blood pressure, consistent with results obtained in *Pkd2* ecKO mice (**MacKay et al., 2020**). In contrast, PC-1 and PC-2 do not contribute to ACh-induced vasodilation, indicating that these proteins are activated by specific stimuli and that their knockout does not cause generalized EC dysfunction. Myogenic tone, depolarization-induced constriction, and passive diameter are also similar in *Pkd1* ecKO, *Pkd2* ecKO, *Pkd1*/*Pkd2* ecKO, and control mouse arteries, illustrating that smooth muscle function is not altered in these knockout mice.

Previous studies have demonstrated that flow activates SK and inwardly rectifying K$^+$ (K$_{ir}$) channels to induce vasodilation (**Ahn et al., 2017**; **Brähler et al., 2009**; **Taylor et al., 2003**). K$_{ir}$2.1 activation was shown to contribute to eNOS phosphorylation, but not to SK channel activation (**Ahn et al., 2017**). The functional significance of PC-1 and PC-2 to the regulation of K$_{ir}$ channels remains to be determined. In arterial membrane potential measurements, we did not verify cell types from which impalements were obtained, but ECs and smooth muscle cells are electrically coupled, particularly in resistance-size arteries (**Garland et al., 2011**). Our data support the following signaling mechanism. Flow-activated PC-1/PC-2-mediated stimulation of eNOS, IK channels, and SK channels produces arterial hyperpolarization, reducing the activity of voltage-dependent (Ca$_V$1.2) channels in smooth muscle cells (**Garland et al., 2011**). The subsequent reduction in intracellular Ca$^{2+}$ concentration in smooth muscle cells produces vasodilation and a decrease in blood pressure (**Jaggar et al., 2000**). These data suggest that PC-1/PC-2 in ECs of other resistance-size arteries may also function to induce vasodilation. Future studies should aim to investigate PC-1/PC-2 expression and function throughout the vasculature.

Virtually all vertebrate cells possess at least one primary cilia, a tiny (~0.2–0.5 µm width, 1–12 µm length) immotile organelle electrically distinct from the cell body (**Kleene and Van Houten, 2014**; **Delling et al., 2013**; **DeCaen et al., 2013**). Cilia generate compartmentalized signaling and maintain an intracellular Ca$^{2+}$ concentration distinct from the cell soma (**Kleene and Van Houten, 2014**; **Delling et al., 2013**; **DeCaen et al., 2013**). A great deal of debate has taken place over whether cells sense external fluid flow through proteins located in primary cilia or the plasma membrane. Primary cilia act as flow sensors in the embryonic node and in kidney collecting duct cells (**Yoshiba et al., 2012**;

*Praetorius and Spring, 2003*). In contrast, flow does not activate $Ca^{2+}$ influx in primary cilia of cultured kidney epithelial cells, kidney thick ascending tubules, embryonic node crown cells, or in the kinocilia of inner ear hair cells (*Delling et al., 2016*). Instead, flow stimulates a cytoplasmic $Ca^{2+}$ signal in the cell body that propagates to cilia to elevate intraciliary $Ca^{2+}$ concentration (*Delling et al., 2016*). Here, flow stimulated plasma membrane cation currents, indicating that flow-sensing takes place in the cell body of ECs.

Physiological functions of PC-1 and the potential involvement of PC-2 in PC-1-mediated signaling mechanisms in endothelial cells were unclear. Whether physical coupling of PC-1 and PC-2 is required for these proteins to traffic to the surface and to activate cellular signaling has also been a matter of debate. Studies that aimed to answer this question primarily used renal epithelial cells or recombinant proteins expressed in cultured cells or *Xenopus* oocytes. Ciliary localization of PC-2 is necessary for flow-sensing in perinodal crown cells and left-right symmetry in mouse embryos (*Yoshiba et al., 2012*). In contrast, other studies demonstrated that a physical interaction between PC-1 and PC-2 is essential for these proteins to traffic to the cell surface and generate plasma membrane currents (*Su et al., 2018*; *Yu et al., 2009*; *Hanaoka et al., 2000*; *Delmas et al., 2004*; *Chapin et al., 2010*; *Gainullin et al., 2015*; *Ha et al., 2020*). It was essential to include an IgҜ-chain secretion sequence into recombinant PC-1 to induce its trafficking and that of associated PC-2 to the plasma membrane in HEK293 cells (*Ha et al., 2020*). We show that when PC-1 and PC-2 are both expressed, they readily traffic to the plasma membrane in ECs. PC-1 and PC-2 protein clusters are present in the EC plasma membrane with more than one-quarter exhibiting nanoscale overlap, supporting their coassembly. Knockout of either PC-1 or PC-2 does not alter the amount of the other polycystin protein, suggesting that the expression of each polycystin is not influenced by that of the other. In contrast, PC-1 or PC-2 knockout prevents surface localization of the other polycystin protein, leading to its intracellular retention, as can be seen in SMLM and immunofluorescence imaging experiments. Flow did not alter the properties of surface PC-1 or PC-2 clusters, or their intercluster distance or overlap, suggesting that flow activates a heteromeric PC-1/PC-2 complex in the plasma membrane. The cluster sizes we report reflect those of the proteins and the antibodies used to label them. Western blotting, immunofluorescence, immunoFRET and SMLM experiments on ECs and arteries of *Pkd1*$^{fl/fl}$, *Pkd2*$^{fl/fl}$, *Pkd1* ecKO, and *Pkd2* ecKO mice have validated the PC-1 and PC-2 antibodies (here and *MacKay et al., 2020*). A small amount of fluorescence was observed in *Pkd1* ecKO and *Pkd2* ecKO cells during SMLM experiments. We use the term 'nonspecific labeling' as an umbrella term to refer to secondary antibodies that may be free, bound to primary antibodies or both. It is for this reason that the comparison between ECs of *Pkd1*$^{fl/fl}$, *Pkd2*$^{fl/fl}$, *Pkd1* ecKO, and *Pkd2* ecKO mice is useful, particularly as the same labeling protocols were performed on ECs of these different mouse models. A previous study showed that PC-2 immunolabeling colocalized with α-tubulin, a ciliary marker, in ECs of embryonic (E15.5) conduit vessels (*AbouAlaiwi et al., 2009*). We did not examine if PC-1 or PC-2 is present in cilia or if flow activates PC-1- or PC-2-dependent currents in the cilia of ECs. Future studies should investigate these possibilities.

PC-1 and PC-2 did not generate currents in the plasma membrane of renal collecting duct cells or following heterologous expression in cell lines (*Liu et al., 2018*; *Arif Pavel et al., 2016*; *Shen et al., 2016*; *Hanaoka et al., 2000*; *Delmas et al., 2004*; *DeCaen et al., 2013*). Knockout of either PC-1 or PC-2 also did not alter plasma membrane currents in inner medullary collecting duct (pIMCD) epithelial cells (*Liu et al., 2018*). Recently, it was demonstrated that plasma membrane PC-1/PC-2 channels are silent but can be measured when a gain-of-function mutation (F604P) is introduced into PC-2 (*Ha et al., 2020*; *Wang et al., 2019*). We show that in the absence of flow, surface PC-1/PC-2 channels generate little current in ECs (*MacKay et al., 2020*). Rather, flow-activates plasma membrane currents in ECs that are similarly attenuated by either PC-1 or PC-2 knockout or by the introduction of peptides that correspond to the C-terminus coiled-coil domains that interact on each protein (here and *MacKay et al., 2020*). Flow-mediated intracellular $Ca^{2+}$ elevations and NO production were attenuated in cultured embryonic aortic ECs from *Pkd1* and Tg737$^{orpk/orpk}$ global knockout mice (*Nauli et al., 2008*). As Tg737$^{orpk/orpk}$ global knockout mice have shorter cilia or no cilia, the authors proposed that PC-1 located in cilia is a flow sensor in embryonic ECs (*Nauli et al., 2008*). In contrast, more recent studies have demonstrated that flow did not elicit $Ca^{2+}$ signaling in primary cilia and homomeric PC-2 channels did not require PC-1 in primary cilia of pIMCD epithelial cells (*Liu et al., 2018*). Our data support the conclusion that flow activates PC-1/PC-2-dependent cation currents in the plasma membrane of ECs.

Multiple different domains in PC-1 and PC-2 physically interact. Several groups have demonstrated that PC-1 and PC-2 couple via their C-terminal coiled-coils (*Qian et al., 1997*; *Zhu et al., 2011*; *Yu et al., 2009*; *Tsiokas et al., 1997*). Recombinant PC-1 and PC-2 lacking coiled-coils can also interact via N-terminal loops (*Babich et al., 2004*; *Feng et al., 2008*). The structure of a PC-1/PC-2 heterotetramer resolved using cryo-EM indicated that a region between TM6 and TM11 of PC-1 interdigitates with PC-2 (*Su et al., 2018*). PC-1 is proposed to act both as a dominant-negative subunit in PC-1/PC-2 channels and to increase $Ca^{2+}$ permeability over that in PC-2 homotetramers (*Su et al., 2018*; *Wang et al., 2019*). Our data suggest that flow stimulates the physical coupling of the coiled-coil domains in PC-1/PC-2 to activate current. Alternatively, the interference peptides may uncouple constitutively bound coiled coils in PC-1/PC-2 heteromers, thereby preventing current activation by flow. It is unlikely that the interference peptides dissolve PC-1/PC-2 into individual subunits as several other domains can interact and the PC-1/PC-2 heteromeric structure was resolved in the absence of the coiled-coils (*Su et al., 2018*; *Feng et al., 2008*). PC-1 has been proposed to act as an atypical G protein due to the presence of a G protein-binding domain located in the C-terminus (*Parnell et al., 1998*). The PC-1 interference peptide we used does not overlap with this G protein-binding domain, which is located between amino acids 4125 and 4143. Thus, G protein signaling by PC-1 may not be involved in flow-mediated PC-1/PC-2-dependent current activation in ECs, although this remains to be determined.

Our data suggest that flow activates $Ca^{2+}$ influx through PC-1/PC-2 channels. Although homomeric PC-2 is a $K^+$- and $Na^+$-permeant channel with low $Ca^{2+}$ permeability, heteromeric assembly of PC-2 with PC-1 increases $Ca^{2+}$ permeability and reduces block by external $Ca^{2+}$, supporting this concept (*Liu et al., 2018*; *Wang et al., 2019*). We did not determine the ionic permeability of flow-activated PC-1/PC-2-dependent currents in ECs (*Wang et al., 2019*). ECs express a wide variety of ion channels, including several other TRPs and $K^+$ channels, making the isolation of a pure PC-1/PC-2-dependent current challenging. PC-1/PC-2 may also interact with other TRP channels. For example, PC-2-containing channels have been suggested to require TRPM3 in primary cilia of cultured renal epithelial cells (*Kleene et al., 2019*). Thus, PC-1/PC-2 channels in ECs may contain other TRP subunits.

Flow may directly or indirectly activate PC-1/PC-2 in ECs. PC-1 is proposed to act as a mechanical sensor and ligand-receptor in cultured kidney epithelial cells (*Hardy and Tsiokas, 2020*; *Zhou, 2009*; *Nauli et al., 2003*). The PC-1 extracellular N-terminus contains several putative adhesion- and ligand-binding sites that may confer mechanosensitivity (*Hughes et al., 1995*; *Burn et al., 1995*; *Qian et al., 2002*). Recent evidence indicates that the C-type lectin domain located in the PC-1 N-terminus activates recombinant PC-1/PC-2 channels, providing one possible activation mechanism for flow (*Ha et al., 2020*). Other mechanosensitive proteins, such as Piezo1 and GPR68, may also stimulate PC-1/PC-2-dependent currents in ECs (*Xu et al., 2018*; *Wang et al., 2016*). Investigating the mechanisms by which flow activates PC-1/PC-2-dependent currents should be a focus of future studies.

ADPKD is typically characterized by the appearance of renal cysts, but patients can develop hypertension prior to any kidney dysfunction, with cardiovascular disease the leading (~50%) cause of death in patients (*Valero et al., 1999*; *Martinez-Vea et al., 2004*; *Torres et al., 2007*; *Gabow, 1990*; *Bergmann et al., 2018*). Hypertension occurs prior to loss of kidney function in more than 60% of patients, with the average age of onset between 30 and 34 years of age (*Chapman et al., 2010*). Our study raises the possibility that hypertension and cardiovascular disease in ADPKD patients may involve altered PC-1/PC-2 function in ECs. Consistent with our data, human ADPKD patients exhibit loss of NO release and a reduction in endothelium-dependent dilation during increased blood flow (*MacKay et al., 2020*; *Lorthioir et al., 2015*). Hypertension in humans is also associated with endothelial dysfunction and attenuated flow-mediated dilation (*Antony et al., 1995*). Conceivably, hypertension may also be associated with dysfunctional PC-1/PC-2 signaling in ECs. Our demonstration that PC-1/PC-2 elicits vasodilation may lead to studies identifying potential dysfunction in patients with hypertension, ADPKD, and other cardiovascular diseases.

In summary, we show that PC-1 regulates arterial contractility through the formation of an interdependent plasma membrane signaling complex with PC-2 in ECs. Flow stimulates PC-1/PC-2-dependent currents in ECs, leading to eNOS, IK channel, and SK channel activation, vasodilation, and a reduction in blood pressure.

## Materials and methods

### Key resources table

| Reagent type (species) or resource | Designation | Source or reference | Identifiers | Additional information |
|---|---|---|---|---|
| Strain, strain background (*Mus musculus*) | *Pkd1^{fl/fl}* | Baltimore PKD Core Center | PMID:15579506 MGI: 3617325 | Mice with *Pkd1* gene flanked by *loxP* regions. |
| Strain, strain background (*M. musculus*) | *Pkd2^{fl/fl}* | Baltimore PKD Core Center | PMID:20862291 MGI: 4843127 | Mice with *Pkd2* gene flanked by *loxP* regions. |
| Strain, strain background (*M. musculus*) | *Pkd1^{fl/fl}* /*Pkd2^{fl/fl}* | This paper | | Mouse line created in-house by mating *Pkd1^{fl/fl}* with *Pkd2^{fl/fl}*. Mice with *Pkd1* and *Pkd2* genes flanked by *loxP* regions. |
| Strain, strain background (*M. musculus*) | *Cdh5*(PAC)-CreERT2 | Cancer Research UK | RRID: MGI: 3848982 | Mice with tamoxifen-inducible Cre recombinase that is expressed specifically in endothelial cells. |
| Strain, strain background (*M. musculus*) | *Pkd1^{fl/fl}*: *Cdh5*(PAC)-CreERT2 | This paper | | Mouse line created in-house by mating *Pkd1^{fl/fl}* with *Cdh5*(PAC)-CreERT2. Mice with inducible endothelial cell-specific deletion of PC-1. |
| Strain, strain background (*M. musculus*) | *Pkd2^{fl/fl}*: *Cdh5*(PAC)-CreERT2 | PMID: 32364494 | | Mouse line created in-house by mating *Pkd2^{fl/fl}* with *Cdh5*(PAC)-CreERT2. Mice with inducible endothelial cell-specific deletion of PC-2. |
| Strain, strain background (*M. musculus*) | *Pkd1^{fl/fl}*/*Pkd2^{fl/fl}*: *Cdh5*(PAC)-CreERT2 | This paper | | Mouse line created in-house by mating *Pkd1^{fl/fl}* *Cdh5*(PAC)-CreERT2 with *Pkd2^{fl/fl}* *Cdh5*(PAC)-CreERT2. Mice with inducible endothelial cell-specific deletion both PC-1 and PC-2. |
| Antibody | Anti-PC-1 (mouse monoclonal) | Baltimore PKD Core | Mouse mAB SF4AZ E3 | IF (1:100) Lattice SIM (1:100) SMLM (1:100) |
| Antibody | Anti-PC-2 (rabbit monoclonal) | Baltimore PKD Core | Rabbit mAB 3374 CT-14/4 | IF (1:100) Lattice SIM (1:100) SMLM (1:100) |
| Antibody | Anti-CD31 (rat monoclonal) | Abcam | Cat. #: Ab7388 RRID: AB_305905 | IF (1:100) Lattice SIM (1:100) SMLM (1:100) |
| Antibody | Anti-PC-1 (H-260) (rabbit polyclonal) | Santa Cruz Biotechnology | Cat. #: sc-25570, RRID: AB_2163357 | N-FRET (1:100) |
| Antibody | Anti-PC-2 (mouse monoclonal) | Santa Cruz Biotechnology | Cat. #: sc-28331, RRID: AB_672377 | N-FRET (1:100) |
| Antibody | Alexa 488-conjugated anti-rat IgG (donkey polyclonal) | Thermo Fisher Scientific | Cat. #: A-21208, RRID: AB_2535794 | IF (1:500) |
| Antibody | Alexa 546-conjugated anti-mouse IgG (donkey polyclonal) | Thermo Fisher Scientific | Cat. #: A-10036, RRID: AB_2534012 | IF (1:500) |
| Antibody | Alexa 488-conjugated anti-mouse IgG (donkey polyclonal) | Thermo Fisher Scientific | Cat. #: A-21202, RRID: AB_141607 | Lattice SIM (1:500) SMLM (1:500) |
| Antibody | Alexa 546-conjugated anti-rabbit IgG (donkey polyclonal) | Thermo Fisher Scientific | Cat. #: A-10040, RRID: AB_2534016 | Lattice SIM (1:500) |
| Antibody | Alexa 647-conjugated anti-rat IgG (goat polyclonal) | Thermo Fisher Scientific | Cat. #: A-21247, RRID: AB_141778 | Lattice SIM (1:500) |
| Antibody | Alexa 555-conjugated anti-rat IgG (goat polyclonal) | Thermo Fisher Scientific | Cat. #: A-21434, RRID: AB_2535855 | SMLM (1:500) |
| Antibody | Alexa 647-conjugated anti-rabbit IgG (donkey polyclonal) | Thermo Fisher Scientific | Cat. #: A-31573, RRID: AB_2536183 | SMLM (1:500) |
| Antibody | Alexa Fluor 488-conjugated anti-mouse (goat polyclonal) | Thermo Fisher Scientific | Cat. #: A-11001, RRID: AB_2534069 | N-FRET (1:100) |
| Antibody | Alexa Fluor 555-conjugated anti-rabbit (donkey polyclonal) | Thermo Fisher Scientific | Cat. #: A-31572, RRID: AB_162543 | N-FRET (1:100) |
| Antibody | Anti-PC-1 (mouse monoclonal) | Santa Cruz Biotechnology | Cat. #: sc-130554, RRID: AB_2163355 | WB (1:100) |

| Reagent type (species) or resource | Designation | Source or reference | Identifiers | Additional information |
|---|---|---|---|---|
| Antibody | Anti-PC-2 (mouse monoclonal) | Santa Cruz Biotechnology | Cat. #: sc-47734, RRID: AB_672380 | WB (1:100) IP (1:20) |
| Antibody | Anti-IK1 (D-5) (mouse monoclonal) | Santa Cruz Biotechnology | Cat. #: sc-365265, RRID: AB_10841432 | WB (1:100) |
| Antibody | Anti-SK3 (rabbit polyclonal) | Abcam | Cat. #: ab28631, RRID: AB_775888 | WB (1:100) |
| Antibody | Anti-eNOS [M221] (mouse monoclonal) | Abcam | Cat. #: ab76198, RRID: AB_1310183 | WB (1:100) |
| Antibody | Anti-TRPV4 (clone 1B2.6) (mouse monoclonal) | Millipore Sigma | Cat. #: MABS466 | WB (1:100) |
| Antibody | Anti-Actin (clone C4) (mouse monoclonal) | MilliporeSigma | Cat. #: MAB1501, RRID: AB_2223041 | WB (1:1000) |
| Antibody | Anti-p-eNOS (rabbit polyclonal) | Cell Signaling Technology | Cat. #: 9571, RRID: AB_329837 | WB (1:100) |
| Antibody | Anti-GPR68 (rabbit polyclonal) | NOVUS Biologicals | Cat. #: NBP2-32747 | WB (1:100) |
| Antibody | Anti-Piezo1 (rabbit polyclonal) | ProteinTech | Cat. #: 15939-1-AP, RRID: AB_2231460 | WB (1:100) |
| Antibody | Mouse IgG control (mouse polyclonal) | MilliporeSigma | Cat. #: 12-371, RRID: AB_145840 | IP (1:20) |
| Peptide, recombinant protein | PC-1 coiled-coil domain peptide | Genscript | FDRLNQATE DVYQLEQQL | 1 µM |
| Peptide, recombinant protein | PC-1 scrambled peptide | Genscript | QLNDLFQTE VAEDLQRYQ | 1 µM |
| Peptide, recombinant protein | PC-2 coiled-coil domain peptide | Genscript | KRREVLGRLL | 1 µM |
| Peptide, recombinant protein | PC-2 scrambled peptide | Genscript | VLKLLRRRGE | 1 µM |
| Commercial assay or kit | Pierce crosslink magnetic IP/co-IP Kit | Thermo Fisher Scientific | Cat. #: 88805 | |
| Other | DAPI stain | Thermo Fisher Scientific | Cat. #: 3571 RRID: AB_2307445 | 1:1000 |

## Animals

All animal studies were performed in accordance with the Institutional Animal Care and Use Committee (IACUC) at the University of Tennessee Health Science Center. $Pkd1^{fl/fl}$ and $Pkd2^{fl/fl}$ mice were obtained from the Baltimore PKD Center (Baltimore, MD). Cdh5(PAC)-CreERT2 mice were a kind gift from Cancer Research UK (*Wang et al., 2010*). $Pkd1^{fl/fl}$ mice were crossed with tamoxifen-inducible EC-specific Cre mice (Cdh5(PAC)-CreERT2, Cancer Research UK) to generate $Pkd1^{fl/fl}$:Cdh5(PAC)-CreERT2 mice. $Pkd2^{fl/fl}$:Cdh5(PAC)-CreERT2 mice were generated as previously described (*MacKay et al., 2020*). $Pkd1^{fl/fl}$ mice were crossed with $Pkd2^{fl/fl}$ mice to produce $Pkd1^{fl/fl}/Pkd2^{fl/fl}$ mice. $Pkd1^{fl/fl}$:Cdh5(PAC)-CreERT2 mice were crossed with $Pkd2^{fl/fl}$:Cdh5(PAC)-CreERT2 mice to generate $Pkd1^{fl/fl}/Pkd2^{fl/fl}$:Cdh5(PAC)-CreERT2 mice. The genotypes of all mice were confirmed using PCR (Transnetyx, Memphis, TN) before use. $Pkd1^{fl/fl}$, $Pkd2^{fl/fl}$ and $Pkd1^{fl/fl}/Pkd2^{fl/fl}$ Cre negative mice were used as controls. All mice (male, 12 weeks of age) were injected with tamoxifen (50 mg/kg, i.p.) once per day for 5 days and studied between 7 and 14 days after the last injection.

## Tissue preparation and EC isolation

Mesenteric artery branches from first to fifth order were cleaned of adventitial tissue and placed into ice-cold physiological saline solution (PSS) that contained (in mM): 112 NaCl, 6 KCl, 24 NaHCO$_3$, 1.8

$CaCl_2$, 1.2 $MgSO_4$, 1.2 $KH_2PO_4$, and 10 glucose, gassed with 21% $O_2$, 5% $CO_2$, and 74% $N_2$ (pH 7.4). ECs were dissociated by introducing EC basal media (Endothelial cell GM MV2, PromoCell) containing 2 mg/ml collagenase type 1 (Worthington Biochemical) into the arterial lumen and left to incubate for 30–40 min at 37°C. EC isolate was placed into EC basal media containing growth supplements (PromoCell) that support EC survival. Primary-cultured ECs were studied within 5 days of isolation.

## Genomic PCR

Genomic DNA was isolated from mesenteric arteries using a Purelink Genomic DNA Kit (Thermo Fisher Scientific). PCR was then performed on an Eppendorf Gradient thermal cycler using the following protocol: 95°C for 2 min, then 35 cycles of 95°C for 30 s, 56°C for 30 s, and 72°C for 30 s. Primer sequences were as follows: *Pkd1*$^{fl/fl}$: GTTATTCGAGGTCGCTAGACCCTATC (forward), GTTA CAGATGAGGCCCAGGGAAAG (reverse), *Pkd1* ecKO: GGTACGAGAGAGAAGTGGTCTCAGGA (forward), GAGATCCCACCGCGGTTTTGCTAGAAGGCA (reverse). PCR products were separated on 1.5% agarose gels.

## Western blotting

Mesenteric artery segments comprising second- to fifth-order vessels were used for Western blotting. Arteries were transferred to an Eppendorf tube containing RIPA Buffer (Sigma-Aldrich: R0278) and protease inhibitor cocktail (Sigma-Aldrich: P8340, 1:100 dilution). For experiments measuring the effect of flow of eNOS phosphorylation, arteries were exposed to flow (15 /dyn) for 5 min at 37°C and then immediately transferred to an Eppendorf tube containing RIPA buffer with protease and phosphatase inhibitor cocktail (1:100 dilution). Arteries were cut into small segments using micro scissors and mechanically broken down using a homogenizer (Argos Technologies: A0001). The lysate was centrifuged at 4°C, 10,000 rpm for 2 min. This process was repeated three times, after which the supernatant was collected. Proteins in lysate were separated on 7.5% SDS-polyacrylamide gels and blotted onto nitrocellulose membranes. Membranes were blocked with 5% milk and incubated with one of the following primary antibodies: PC-1 (Santa Cruz Biotechnology), PC-2 (Santa Cruz Biotechnology), Piezo1 (ProteinTech), GPR68 (NOVUS), SK3 (Sigma-Aldrich [P0608]), IK (Alomone), TRPV4 (MilliporeSigma), eNOS (Abcam), or actin (MilliporeSigma) overnight at 4°C. Membranes were washed and incubated with horseradish peroxidase-conjugated secondary antibodies at room temperature. Protein bands were imaged using a ChemiDoc Touch Imaging System (Bio-Rad), quantified using ImageJ software, and normalized to actin.

## Laser-scanning confocal microscopy

Arteries were cut open longitudinally and fixed with 4% paraformaldehyde in phosphate-buffered saline (PBS) for 1 hr. Following a wash in PBS, arteries were permeabilized with 0.2% Triton X-100 and blocked with 3% BSA +5% serum for 1 hr at room temperature. For en face imaging experiments, arteries were incubated overnight with anti-PC-1 monoclonal primary antibody (E3 5F4A2, Baltimore PKD Center) and anti-CD31 primary monoclonal antibody (Abcam 7388) at 4°C. Arteries were then incubated with Alexa Fluor 488 donkey anti-rat, Alexa Fluor 546 donkey anti-mouse secondary antibodies (1:500; Molecular Probes), and 4',6-diamidino-2-phenylindole, dihydrochloride (DAPI) (1:1000; Thermo Fisher Scientific) for 1 hr at room temperature. After washing with PBS, arteries were mounted in 80% glycerol solution. DAPI, Alexa 488, and Alexa 546 were excited at 405, 488, and 561 nm with emission collected at ≤460 nm and ≥500 nm, respectively, using a Zeiss LSM 710 laser-scanning confocal microscope.

## Patch-clamp electrophysiology

The conventional whole-cell configuration was used to measure steady-state currents in primary-cultured ECs at a holding potential of –60 mV. Cells used label with CD31 (*Figure 4E*), respond to flow and produce SK and IK currents (*MacKay et al., 2020*), consistent with ECs. For experiments using physiological ionic gradients, the bath solution contained (in mM): NaCl 134, KCl 6, HEPES 10, $MgCl_2$ 1, $CaCl_2$ 2, and glucose 10 (pH 7.4). The $Ca^{2+}$-free bath solution was the same composition as bath solution except $Ca^{2+}$ was omitted and 1 mM EGTA was included. The pipette solution contained (in mM): K aspartate 110, KCl 30, HEPES 10, glucose 10, EGTA 1, Mg-ATP 1, and Na-GTP 0.2, with total $MgCl_2$ and $CaCl_2$ adjusted to give free concentrations of 1 mM $Mg^{2+}$ and 200 nM $Ca^{2+}$, respectively

(pH 7.2). For experiments measuring $I_{Cat}$, the bath solution contained (in mM): Na aspartate 135, NaCl 5, HEPES 10, glucose 10, and MgCl$_2$ 1 (pH 7.4). A low Na$^+$ bath solution used when measuring $I_{Cat}$ was (in mM): NMDG-asp 135, NaCl 5, HEPES 10, glucose 10, and MgCl$_2$ 1 (pH 7.4). The pipette solution contained (in mM): Na aspartate 135, NaCl 5, HEPES 10, glucose 10, EGTA 1, Mg-ATP 1, and Na-GTP 0.2, with total Mg$^{2+}$ and Ca$^{2+}$ adjusted to give free concentrations of 1 mM and 200 nM, respectively (pH 7.2). PC-1 and PC-2 coiled-coil domain and corresponding scrambled peptides were custom-made (Genscript) and added to the pipette solution immediately before use. Free Mg$^{2+}$ and Ca$^{2+}$ were calculated using WebmaxC Standard. The osmolarity of solutions was measured using a Wescor 5500 Vapor Pressure Osmometer (Logan, UT). Currents were filtered at 1 kHz and digitized at 5 kHz using an Axopatch 200B amplifier and Clampex 10.4 (Molecular Devices). Offline analysis was performed using (Clampfit 10.4). The flow-activated transient inward current was measured at its peak in each cell. The steady-state inward current was the average of at least 45 s of contiguous data.

## Pressurized artery membrane potential measurements

Membrane potential was measured by inserting sharp glass microelectrodes (50–90 MΩ) filled with 3 M KCl into the adventitial side of pressurized third- and fourth-order mesenteric arteries. Membrane potential was recorded using a WPI FD223a amplifier and digitized using a MiniDigi 1A USB interface, pClamp 9.2 software (Axon Instruments) and a personal computer. Criteria for successful intracellular impalements were a sharp negative deflection in potential on insertion, stable voltage for at least 1 min after entry, a sharp positive voltage deflection on exit from the recorded cell, and a <10% change in tip resistance after impalement.

## Pressurized artery myography

Experiments were performed using isolated third- and fourth-order mesenteric arteries using PSS gassed with 21% O$_2$/5% CO$_2$/74% N$_2$ (pH 7.4). Arterial segments 1–2 mm in length were cannulated at each end in a perfusion chamber (Living Systems Instrumentation) continuously perfused with PSS and maintained at 37°C. Intravascular pressure was altered using a Servo pump model PS-200-P (Living Systems) and monitored using pressure transducers. Following the development of stable myogenic tone, intraluminal flow was introduced using a P720 peristaltic pump (Instech). The intraluminal flow rate required to apply a specific amount of shear stress to each artery was calculated using internal diameter. Arterial diameter was measured at 1 Hz using a CCD camera attached to a Nikon TS100-F microscope and the automatic edge-detection function of IonWizard software (Ionoptix). Myogenic tone was calculated as: $100 \times (1 - D_{active}/D_{passive})$ where $D_{active}$ is active arterial diameter and $D_{passive}$ is the diameter determined in the presence of Ca$^{2+}$-free PSS supplemented with 5 mM EGTA.

## Telemetric blood pressure and locomotion measurements

Telemetric blood pressure recordings were performed at the University of Tennessee Health Science Center. Briefly, transmitters (PA-C10, Data Sciences International) were implanted subcutaneously into anesthetized mice, with the sensing electrode placed in the aorta via the left carotid artery. Mice were allowed to recover for 7–10 days. Blood pressure was then recorded every 20 s for 5 days prior to tamoxifen injection and again for the entire time period between 7 and 22 days after the last tamoxifen injection (50 mg/kg per day for 5 consecutive days, i.p) using a PhysioTel Digital telemetry platform (Data Sciences International). Dataquest A.R.T. software was used to acquire and analyze data.

## Kidney histology

Kidney sections were stained with H&E and examined by Probetex, Inc (San Antonio, TX). Briefly, image analysis was performed to measure the glomerular size and tubular cross-sectional diameter. The glomerular size was measured by tracing the circumference of 25 random glomeruli and surface area was calculated using the polygonal area tool of Image-Pro 4.5 image analysis software calibrated to a stage micrometer. Tubular size was measured using the linear length tool of Image-Pro 4.5 imaging software. The tracing tool was applied at the diameter of cross-sectional profiles of 5 proximal tubules/image (total of 25/section). Glomerular and tubular images were calibrated to a stage micrometer and data was transferred to an Excel spreadsheet and statistical analysis was performed by Excel analysis pack.

## Co-immunoprecipitation

Mesenteric artery segments comprising second- to fifth-order vessels were used. Arteries were transferred to an Eppendorf tube containing lysis buffer and protease inhibitor cocktail (Sigma-Aldrich: P8340, 1 in 100 dilution). Arteries were cut into small segments using micro scissors and broken down using a mechanical homogenizer (Argos Technologies: A0001). Arterial lysate was centrifuged for 2 min at 10,000 rpm and at 4°C. This process was repeated three times, after which the supernatant was collected. Proteins were pulled down from arterial lysate using a Pierce crosslink Magnetic IP/coIP kit (Thermo Fisher Scientific) as per the manufacturer's instructions. Samples were incubated with PC-2 antibody (1:20, Santa Cruz Biotechnology) that was covalently bound to protein A/G Magnetic Beads (Pierce) overnight at 4°C. Following washing and elution, immunoprecipitates were analyzed using Western blotting.

## Immunofluorescence resonance energy transfer (immunoFRET) imaging

Primary-cultured mesenteric artery ECs were fixed with paraformaldehyde and permeabilized with 0.1% Triton X-100 for 2 min at room temperature. After blocking with 5% bovine serum albumin (BSA), the cells were treated overnight at 4°C with anti PC-1 (Rabbit polyclonal, Santa Cruz Biotechnology, sc-25570, 1:100 dilution, RRID:AB_2163357) and anti PC-2 (mouse monoclonal, Santa Cruz Biotechnology, sc-28331, 1:100 dilution, RRID:AB_672377) antibodies in PBS containing 5% BSA. After a wash, cells were incubated for 1 hr at 37°C with secondary antibodies: goat anti-Mouse Alexa Fluor 488 (Thermo Fisher Scientific, A-11001) and donkey anti-Rabbit Alexa Fluor 555 (Thermo Fisher Scientific, A-31572). Coverslips were then washed and mounted on glass slides. Fluorescence images were acquired using a Zeiss 710 laser-scanning confocal microscope. Alexa 488 and Alexa 546 were excited at 488 and 543 nm and emission collected at 505–530 and >560 nm, respectively. Images were acquired using a z-resolution of ~1 μm. Images were background-subtracted and normalized FRET (N-FRET) was calculated on a pixel-by-pixel basis for the entire image and in regions of interest (within the boundaries of the cell) using the Xia method (*Dimmeler et al., 1999*) and Zeiss LSM FRET Macro tool version 2.5 as previously described (*Delling et al., 2013*).

## Lattice structured illumination microscopy (Lattice-SIM)

Arteries were cut open longitudinally and fixed with 4% paraformaldehyde in PBS for 1 hr. Following a wash in PBS, arteries were permeabilized with 0.2% Triton X-100 and blocked with 3% BSA +5% serum for 1 hr at room temperature. Arteries were incubated overnight with anti-PC-1 monoclonal primary antibody (E3 5F4A2, Baltimore PKD Center), anti-PC-2 monoclonal antibody (3374 CT-1 414, Baltimore PKD Center), and anti-CD31 primary monoclonal antibody (Abcam 7388) at 4°C. Arteries were then incubated with Alexa Fluor 488 donkey anti-mouse, Alexa Fluor 546 donkey anti-rabbit secondary antibodies, and Alexa Fluor 647 goat anti-rat secondary antibodies (1:500; Molecular Probes) for 1 hr at room temperature. After washing with PBS, arteries were mounted in 80% glycerol solution. Lattice-SIM images were acquired on a Zeiss Elyra 7 AxioObserver microscope equipped with a 63× Plan-Apochromat (NA 1.46) oil immersion lens and an sCMOS camera. Lattice-SIM reconstruction was performed using the SIM processing Tool of Zeiss ZEN Black 3.0 SR software. Colocalization analysis of Lattice-SIM data was performed using Pearson's and Mander's coefficients.

## Single-molecule localization microscopy

Primary-cultured mesenteric artery ECs were seeded onto 35 mm glass-bottom dishes (MatTeK Corp). Cells were fixed in 4% paraformaldehyde, permeabilized with 0.5% Triton X-100 PBS solution and then blocked with 5% BSA. Cells were immunolabelled using primary antibodies to PC-1 (Baltimore PKD Center), PC-2 (Baltimore PKD Center), and CD31 (Abcam 7388) overnight at 4°C. Alexa Fluor 488 donkey anti-mouse, Alexa Fluor 555 goat anti-rat, and Alexa Fluor 647 donkey anti-rabbit secondary antibodies were used for detection. An oxygen scavenging, thiol-based photo-switching imaging buffer was used (GLOX: 50% glucose, 10% PBS, 24 mg/ml glucose oxidase, and 12.6 mg/ml catalase supplemented with a reducing agent (cysteamine hydrochloride-MEA) at 100 mM, pH 7.8). Cells were imaged using a super-resolution Zeiss Elyra 7 microscope using 488 nm (500 mW), 561 nm (500 mW), and 642 nm (500 mW) lasers. A 63× Plan-Apochromat (NA 1.46) oil immersion lens and a CMOS camera were used to acquire images. The camera was run in frame-transfer mode at a rate of 100 Hz

(30 ms exposure time). Fluorescence was detected using TIRF mode with emission band-pass filters of 550–650 and 660–760 nm.

Localization precision was calculated using:

$$\sigma_{x,y}^2 = [(s^2 + q^2/12)/N] + [(8\pi s^4 b^2)/(q^2 N^2)]$$

where $\sigma_{x,y}$ is the localization precision of a fluorescent probe in lateral dimensions, s is the standard deviation of the point-spread function, N is the total number of photons gathered, q is the pixel size in the image space, and b is the background noise per pixel. The precision of localization is proportional to DLR/$\sqrt{N}$, where DLR is the diffraction-limited resolution of a fluorophore and N is the average number of detected photons per switching event, assuming the point-spread functions are Gaussian. PC-1 and PC-2 localization were reconstructed from 35,000 to 40,000 images based on fitting signals to a Gaussian function and taking into account a point spread function calculated using a standardized 40 nm bead slide (ZEN Black software, Zeiss). The first 5000 frames were excluded from the reconstruction to account for time to stabilize photo-switching of the probes. Software drift correction was applied using a model-based cross-correlation.

## Statistical analysis

OriginLab and GraphPad InStat software were used for statistical analyses. Values are expressed as mean ± SEM. Student's t-test was used for comparing paired and unpaired data from two populations and ANOVA with Holm-Sidak post hoc test was used for multiple group comparisons. $p < 0.05$ was considered significant. Power analysis was performed to verify that the sample size gave a value >0.8 if p was >0.05. Kidney histology and blood pressure experiments were all done single-blind, wherein the person performing the experiments and analysis of the results was not aware of the mouse genotype.

## Acknowledgements

This work was supported by NIH/NHLBI grants HL133256, HL137745, HL155180, HL155186 (to JHJ), and HL19134-46 (to KUM), and an American Heart Association (AHA) Postdoctoral Fellowship (20POST35210200), and Career Development Award R073037556 (to CM). The authors thank Dr. Simon Bulley for initial breeding of mouse lines and Dr. Manuel Navedo (UC, Davis) for help with ImageJ software. The authors thank the Advanced Imaging Core at the University of Tennessee Health Science Center for technical assistance during super-resolution imaging experiments.

## Additional information

### Funding

| Funder | Grant reference number | Author |
| --- | --- | --- |
| National Heart, Lung, and Blood Institute | HL133256 | Jonathan H Jaggar |
| National Heart, Lung, and Blood Institute | HL137745 | Jonathan H Jaggar |
| National Heart, Lung, and Blood Institute | HL155180 | Jonathan H Jaggar |
| National Heart, Lung, and Blood Institute | Hl155186 | Jonathan H Jaggar |
| National Heart, Lung, and Blood Institute | HL19134 | Kafait U Malik |
| American Heart Association | 20POST35210200 | Charles E MacKay |
| American Heart Association | 855946 | Charles E MacKay |

| Funder | Grant reference number | Author |
| --- | --- | --- |
| National Heart, Lung, and Blood Institute | HL149662 | M Dennis Leo |

The funders had no role in study design, data collection and interpretation, or the decision to submit the work for publication.

## Author contributions

Charles E MacKay, Conceptualization, Data curation, Formal analysis, Funding acquisition, Investigation, Methodology, Writing - review and editing; Miranda Floen, M Dennis Leo, Raquibul Hasan, Carlos Fernández-Peña, Purnima Singh, Data curation, Formal analysis, Writing - review and editing; Tessa AC Garrud, Data curation, Formal analysis; Kafait U Malik, Funding acquisition, Supervision, Writing - review and editing; Jonathan H Jaggar, Conceptualization, Funding acquisition, Project administration, Resources, Supervision, Writing - original draft, Writing - review and editing

## Author ORCIDs

Charles E MacKay (ID) http://orcid.org/0000-0002-2875-0677
Jonathan H Jaggar (ID) http://orcid.org/0000-0003-1505-3335

## Ethics

This study was performed in strict accordance with the recommendations in the Guide for the Care and Use of Laboratory Animals of the National Institutes of Health. All of the animals were handled according to an approved institutional animal care and use committee (IACUC) protocol (#20-0168) of the University of Tennessee. All surgery was performed under anesthesia, and every effort was made to minimize suffering.

## Decision letter and Author response

Decision letter https://doi.org/10.7554/eLife.74765.sa1
Author response https://doi.org/10.7554/eLife.74765.sa2

# Additional files

## Supplementary files

- Transparent reporting form
- Source data 1. Unlabeled source data.
- Source data 2. Labeled source data.

## Data availability

All data generated or analyzed during this study are included in the manuscript.

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
