## [Editor Report]

This is a very significant study that enhances our understanding of a mysterious ion channel and its function in vascular function.

---

## [Decision Letter]

**Decision letter after peer review:**

Thank you for submitting your article "A plasma membrane-localized polycystin-1/polycystin-2 complex in endothelial cells elicits vasodilation" for consideration by *eLife*. Your article has been reviewed by 3 peer reviewers, one of whom is a member of our Board of Reviewing Editors, and the evaluation has been overseen by Kenton Swartz as the Senior Editor. The following individual involved in review of your submission has agreed to reveal their identity: William Jackson (Reviewer #3).

Essential revisions:

1) A key finding from the current study is the suggestion that interactions between PC-1 and PC-2 via coiled-coil domains are required for activation of inward current by flow. However, the authors did not show evidence, via fluorescence imaging or otherwise (e.g., coIP), that peptides generated to disrupt this interaction actually do so. Does treatment with the coiled-coil domain peptides cause a shift in the PC-1-to-PC-2 distance (using TIRF-SMLM as in Figure 5)? How do you know that the peptides you introduced reduced physical coupling of PC-1 and PC-2 rather than simply acting to block the ion channel? Please show appropriate controls using scrambled peptides. Further, negative controls for immunoFRET (e.g., use of cells from KO models, or FRET between PC-1 and another membrane protein) should be provided to demonstrate the specificity of this technique for PC-1/PC-2 interactions in endothelial cells. Are the currents shown in 6D and 6E for control and with CC peptides for different cells? Please clarify.

2) The conclusions of the manuscript state that endothelial PC-1/PC-2 complexes control arterial contractility through ca^2+^-dependent activation of eNOS and SK channels. However, neither increases in intracellular ca^2+^ concentrations nor NO production were directly measured. Can PC-1/PC-2 mediated ca^2+^ Sparklets be resolved? The authors argue that G-protein signaling is not involved – Can this be tested directly using Gq antagonists or other approaches?

3) What effect does apamin alone and TRAM-34 alone have on the dilation? Authors should provide experiments where for each inhibitor (Apamin and Tram34) are added to the prep first before adding the other two inhibitors. Further, is the hyperpolarization induced by flow blocked by apamin, Tram-34 or their combination? Were there any differences between freshly isolated cells and primary cultured cells? How did you verify that the cells you were using were indeed endothelial cells?

4) Why were different antibodies used for imaging than for Westerns? How do you know that they identified the same proteins? It is not clear what steps were taken to document the specificity of all PC-1 and PC-2 Abs. Especially, that the bulk of interactions studies rely on these Abs and that some staining remains in the KO conditions. In fact, the authors conclude that PC-1/PC-2 clusters in KO cells in SMLM experiments are likely due to non-specific antibody binding, which raises the question as to the meaning of cluster size data. Considering that the approach relies on fluorophore-tagged antibodies, which cannot be assumed to be in 1:1 stoichiometry with proteins of interest, how relevant is cluster size?

5) Based on data shown in Figure 1, the authors conclude that there is a reduction in inward current with flow. Since the applied technique measures total current, couldn't this result also reflect an increase in outward current (e.g., K^+^) due to flow that depends on the presence of ca^2+^? Related to these data, the magnitude of initial flow-induced transient current was quite variable (~8 – ~45 pA). Was this due to differences in cell size? The authors should express data from current recordings as current density (pA/pF). How does the patch clamp data in Fig6D relate to that in Figure 1D where one shows a transient inward current followed by a major reduction in current while the other shows a large monovalent current stimulated by flow? What is the rationale for the use of these two protocols? Can the authors run voltage steps to determine I/V relationships of PC-2?

6) In the section titled: "Endothelial cell PC-1 contributes to flow-mediated arterial hyperpolarization" – When you state arterial membrane potential is this the membrane potential of vascular smooth muscle cells? If so how did you verify which cell type you were recording from? How do you know that the short-term culture did not affect expression and distribution of PC-1 and PC-2 or other ion channels in your cells? Please also define what the flow rate was and what the estimated shear stress was in this section.

Other additional revisions:

1) Data by others suggest that KIR2.1 channels are involved in flow-mediated dilation – what effect does Ba2+ have in your hands? Are KIR2.1 channels also involved? What about other endothelial cell ion channels implicated in flow-mediated dilation? how the involvement of these other channels fits into an integrated scheme with PC-1 and PC-2 should be addressed in the Discussion.

2) Endothelium-specific deletion of PC-1 increased blood pressure, implying that the proposed role for PC-1 is generally applicable to the resistance arterial network; yet here, only small mesenteric vessels were studied. Given the known heterogeneity in the regulation of vascular tone by sheer stress among different arterial beds, is the identified role of PC-1 observed outside of the mesenteric circulation?

3) For microscopy studies – Were these vessels pinned out as flat sheets or simply cut open and then fixed? Please clarify.

4) Telemetric blood pressure and locomotion measurements – How long after the telemeter implantation were mice studied? Please clarify.

5) Why were only male mice used? Please justify.

6) Introduction, page 2 – "…with much of this knowledge derived from experiments studying recombinant proteins and cultured cells." It would be helpful to readers to include references for this statement.

7) What percentage of patients with ADPKD have clinical hypertension? Discussion of this may add to significance.

8) Results, 1st section, page 3 – groups are labeled "Pkd1 ecKO" in Figure 1A, B, but aren't defined as such until later in the narrative.

9) In many graphs, there are data points that fall outside the y-axis scales that are provided. Please correct these.

10) In addition to flow, can the authors speculate on potential ligands that activate PC-1 that may be relevant to the vasculature and blood flow regulation?

11) Does flow alters FRET between PC1/PC2.

12) In Figure 6A actin controls are not equivalent between conditions. All blots (including Figure 6A) should be quantified, normalized to controls, reported as scatter plots and statistically analyzed.

13) Bottom of Page 6. Figure 6C,D should in fact refer to Figure 6B,C. Please change?

---

## [Author Response]

Essential revisions:1) A key finding from the current study is the suggestion that interactions between PC-1 and PC-2 via coiled-coil domains are required for activation of inward current by flow. However, the authors did not show evidence, via fluorescence imaging or otherwise (e.g., coIP), that peptides generated to disrupt this interaction actually do so. Does treatment with the coiled-coil domain peptides cause a shift in the PC-1-to-PC-2 distance (using TIRF-SMLM as in Figure 5)? How do you know that the peptides you introduced reduced physical coupling of PC-1 and PC-2 rather than simply acting to block the ion channel? Please show appropriate controls using scrambled peptides. Further, negative controls for immunoFRET (e.g., use of cells from KO models, or FRET between PC-1 and another membrane protein) should be provided to demonstrate the specificity of this technique for PC-1/PC-2 interactions in endothelial cells. Are the currents shown in 6D and 6E for control and with CC peptides for different cells? Please clarify.

This is an excellent question. To clarify this issue, we have performed new experiments using scrambled PC-1 and PC-2 peptides. We now show that scrambled sequences of the coiled-coil domain peptides do not inhibit flow-activated non-selective cation currents in endothelial cells (new Figure 6E-G and Figure 6 —figure supplement 2). In contrast, peptides corresponding to the coiled-coil domains in either PC-1 or PC-2 similarly inhibit flow-activated cation currents. We consider it highly unlikely that the coiled-coil domain peptides physically separate PC-1 and PC-2 subunits, as we have now discussed in the manuscript. Multiple different domains in PC-1 and PC-2 physically interact to form PC-1/PC-2 heteromers. The C-terminal coiled-coils are only one region where PC-1 and PC-2 interact (Qian et al., Nat. Genet. 1997; Zhu et al., PNAS 2011; Yu et al., PNAS 2009; Tsiokas et al., PNAS 1997). PC-1 and PC-2 proteins that lack coiled-coils interact via N-terminal loops (Babich et al., JBC 2004; Feng et al., JBC 2008). The structure of an N- and C-terminus-deficient PC-1/PC-2 heterotetramer has also been resolved using cryo-EM. In this study, it was found that a region between TM6 and TM11 of PC-1 interdigitates with PC-2 (Su et al., Science 2018). Our data with the peptides suggest that coiled-coil domain coupling is required for flow to activate PC-1/PC-2.

As requested, we also provide new data performed using immunoFRET in *Pkd1* ecKO and *Pkd2* ecKO endothelial cells. These new data provide controls for the other immunoFRET results (Figure 4B, C; Figure 4 —figure supplement 1A, B).

Yes, the data obtained in figure 6E-G and Figure 6 —figure supplement 2 are collected from different endothelial cells, as we now state in the manuscript.

2) The conclusions of the manuscript state that endothelial PC-1/PC-2 complexes control arterial contractility through ca^2+^-dependent activation of eNOS and SK channels. However, neither increases in intracellular ca^2+^ concentrations nor NO production were directly measured. Can PC-1/PC-2 mediated ca^2+^ Sparklets be resolved? The authors argue that G-protein signaling is not involved – Can this be tested directly using Gq antagonists or other approaches?

We have performed new experiments and now show that flow stimulates eNOS phosphorylation in *Pkd1^fl/fl^* arteries and that this effect is attenuated in arteries of *Pkd1* ecKO mice (new Figure 2I, J). These data are similar to those we have previously published for *Pkd2^fl/fl^* and *Pkd2* ecKO arteries, providing further support for our conclusion that PC-1 and PC-2 are interdependent for flow-mediated vasodilation. Whether PC-1/PC-2 activates eNOS, IK channels and SK channels through local or global ca^2+^ signaling is unclear. We consider answering this question to be beyond the scope of the current manuscript and suitable for a future study.

3) What effect does apamin alone and TRAM-34 alone have on the dilation? Authors should provide experiments where for each inhibitor (Apamin and Tram34) are added to the prep first before adding the other two inhibitors. Further, is the hyperpolarization induced by flow blocked by apamin, Tram-34 or their combination? Were there any differences between freshly isolated cells and primary cultured cells? How did you verify that the cells you were using were indeed endothelial cells?

We agree that further investigation of the individual contributions of NOS, SK channels and IK channels to flow-mediated vasodilation caused by PC-1/PC-2 is appropriate. As such, we have performed new experiments. These new data show that L-NNA, Tram-34 and apamin each reduce flow-mediated vasodilation in *Pkd1^fl/fl^* arteries and that these effects are attenuated in arteries of *Pkd1* ecKO mice (new Figure 2E-H). It is well established that NOS, IK channel and SK channel activation in endothelial cells produces arterial hyperpolarization, which leads to vasodilation. We show that flow-mediated arterial hyperpolarization is attenuated in arteries of *Pkd1* ecKO mice, consistent with other data in our manuscript demonstrating that PC-1 stimulates eNOS, IK channels and SK channels. We have now included statements in the manuscript to strengthen this point. Data obtained in primary cultured cells, fresh-isolated arteries and in vivo measurements all produced data consistent with our conclusions. Immunofluorescence, lattice-SIM and SMLM experiments were performed on cells which labeled for CD31, an endothelial cell-specific marker, as is stated in the manuscript. Cells used for patch-clamp experiments label with CD31 (Figure 4F), respond to flow and produce SK and IK currents (6), consistent with endothelial cells. This statement is now included in the Methods.

4) Why were different antibodies used for imaging than for Westerns? How do you know that they identified the same proteins? It is not clear what steps were taken to document the specificity of all PC-1 and PC-2 Abs. Especially, that the bulk of interactions studies rely on these Abs and that some staining remains in the KO conditions. In fact, the authors conclude that PC-1/PC-2 clusters in KO cells in SMLM experiments are likely due to non-specific antibody binding, which raises the question as to the meaning of cluster size data. Considering that the approach relies on fluorophore-tagged antibodies, which cannot be assumed to be in 1:1 stoichiometry with proteins of interest, how relevant is cluster size?

We thank the reviewers for these suggestions. Proteins on Western blots are denatured, whereas proteins in fixed cells are cross-linked. Often, antibodies are better at detecting either denatured or cross-linked proteins. It is for this reason that we used different antibodies for Westerns and immunofluorescence. Western blots of arteries from *Pkd1* ecKO and *Pkd2* ecKO mice validated PC-1 and PC-2 antibodies (here and Mackay et al. *eLife* 2020). Immunofluorescence, immunoFRET and SMLM experiments validated PC-1 and PC-2 antibodies in immunofluorescence experiments performed on *Pkd1* ecKO and *Pkd2* ecKO mice (here and Mackay et al. *eLife* 2020). This information is now stated in the manuscript. A small amount of secondary antibody fluorescence is present in *Pkd1* ecKO and *Pkd2* ecKO cells when performing SMLM, a highly sensitive imaging technique. We used “non-specific labeling” as an umbrella term to refer to secondary antibodies that may be either free or bound to primary antibodies. It is for this reason that the comparison between endothelial cells of *Pkd1^fl/fl^*, *Pkd2^fl/fl^*, *Pkd1* ecKO and *Pkd2* ecKO mice is useful. The same labeling procedures were used in endothelial cells of these different mouse models, making the comparison of cluster sizes relevant. We recognize your comment and now include text stating that the size of the PC-1 and PC-2 clusters we report is the size of both the proteins and the antibodies.

5) Based on data shown in Figure 1, the authors conclude that there is a reduction in inward current with flow. Since the applied technique measures total current, couldn't this result also reflect an increase in outward current (e.g., K^+^) due to flow that depends on the presence of ca^2+^? Related to these data, the magnitude of initial flow-induced transient current was quite variable (~8 – ~45 pA). Was this due to differences in cell size? The authors should express data from current recordings as current density (pA/pF). How does the patch clamp data in Fig6D relate to that in Figure 1D where one shows a transient inward current followed by a major reduction in current while the other shows a large monovalent current stimulated by flow? What is the rationale for the use of these two protocols? Can the authors run voltage steps to determine I/V relationships of PC-2?

We agree that presenting the mean data as current density is useful. As such, we now express all patch-clamp mean data as current density (pA/pF). This reanalysis did not change any of our conclusions. We also agree that in figure 1, the results reflect a flow-activated increase in K^+^ current, as it is partially inhibited by apamin/tram-34 (Mackay et al. *eLife* 2020). We use the term “a reduction in inward current” because the entire current range in these experiments is negative of 0 pA and is therefore, inward. In figure 1, our rationale was to compare the contribution of PC-1 to flow-activated currents in physiological ionic gradients. We have performed similar experiments using *Pkd2^fl/fl^* and *Pkd2* ecKO mice, allowing comparison of these results in the different knockout mouse models (Mackay et al., *eLife* 2020). In figure 6E-G and Figure 6 —figure supplement 2, as we aimed to test the hypothesis that coiled-coil domains in PC-1 and PC-2 may be physiologically relevant. As such, our rationale was to isolate non-selective cation currents to which PC-1 and PC-2 contribute. We have strengthened the rationale for these different experimental conditions by expanding discussion in the manuscript. We have also discussed that the flow-activated reduction in inward current observed when using physiological ionic gradients reflects the activation of K^+^ current due to SK and IK channels, consistent with our previous study and those obtained here when using myography (Mackay et al. *eLife* 2020). Endothelial cells express several different non-selective cation channels, including multiple members of the TRP family. The experiment you suggest would record current from multiple different channel types and would be unlikely to record a pure PC-2 current.

6) In the section titled: "Endothelial cell PC-1 contributes to flow-mediated arterial hyperpolarization" – When you state arterial membrane potential is this the membrane potential of vascular smooth muscle cells? If so how did you verify which cell type you were recording from? How do you know that the short-term culture did not affect expression and distribution of PC-1 and PC-2 or other ion channels in your cells? Please also define what the flow rate was and what the estimated shear stress was in this section.

As suggested, we now provide additional explanation of the microelectrode impalement experiments in the Discussion. Endothelial and smooth muscle cells are electrically coupled. We did not verify from which cell types the membrane potential measurements were obtained. Therefore, we use the term “arterial potential”. We did not determine whether short-term culture affects expression or distribution of channels in endothelial cells. There are both advantages and disadvantages to studying fresh-isolated and primary-cultured endothelial cells. Fresh-isolated cells have very recently been exposed to enzymes, which could reduce surface protein abundance and alter distribution, which will alter results obtained. Primary culture would allow recovery of protein expression and distribution. Our data obtained using primary-cultured endothelial cells, fresh-isolated arteries and in vivo telemetry are consistent with the conclusion that flow activates PC-1/PC-2 clusters in endothelial cells, leading to ca^2+^ influx which activates SK, IK and eNOS, eliciting membrane hyperpolarization, vasodilation and a reduction in blood pressure. As requested, we have included the shear stress (15 dyn/cm^2^) applied to these arteries in the legend. During experimentation, the flow rate applied to induce a required amount of shear stress to an artery was calculated from its internal diameter. We have now included this information in the Methods.

Other additional revisions:1) Data by others suggest that KIR2.1 channels are involved in flow-mediated dilation – what effect does Ba2+ have in your hands? Are KIR2.1 channels also involved? What about other endothelial cell ion channels implicated in flow-mediated dilation? how the involvement of these other channels fits into an integrated scheme with PC-1 and PC-2 should be addressed in the Discussion.

As suggested, we have now discussed previous studies demonstrating that SK and K_ir_ channels contribute to flow-mediated vasodilation. The goal of our study was to investigate the regulation of arterial contractility by PC-1 in endothelial cells. We provide a mechanistic link to eNOS, IK and SK channels. It is beyond the scope of this manuscript to study all previously proposed mechanisms for flow in the context of PC-1 and PC-2. Future studies will be designed to address other mechanisms by which PC-1/PC-2 may signal to provide an integrated scheme.

2) Endothelium-specific deletion of PC-1 increased blood pressure, implying that the proposed role for PC-1 is generally applicable to the resistance arterial network; yet here, only small mesenteric vessels were studied. Given the known heterogeneity in the regulation of vascular tone by sheer stress among different arterial beds, is the identified role of PC-1 observed outside of the mesenteric circulation?

We agree that the blood pressure phenotype in *Pkd1* ecKO mice suggest that flow-activates PC-1 in endothelial cells of other vascular beds to induce vasodilation. We have now discussed this concept in the manuscript.

3) For microscopy studies – Were these vessels pinned out as flat sheets or simply cut open and then fixed? Please clarify.

They were cut open and then fixed. This has now been clarified in the Methods.

4) Telemetric blood pressure and locomotion measurements – How long after the telemeter implantation were mice studied? Please clarify.

This information has now been included in the Methods.

5) Why were only male mice used? Please justify.

Only male mice were used as it was beyond the scope of this study to determine whether sex-specific differences occur in PC-1 and PC-2 signaling in endothelial cells. This would be done in a future study designed to examine sexual dimorphism.

6) Introduction, page 2 – "…with much of this knowledge derived from experiments studying recombinant proteins and cultured cells." It would be helpful to readers to include references for this statement.

We agree and now have included references.

7) What percentage of patients with ADPKD have clinical hypertension? Discussion of this may add to significance.

Hypertension occurs prior to loss of kidney function in more than 60 % of patients, with the average age of onset between 30 and 34 years of age. We have now stated this information in the manuscript and cited references.

8) Results, 1st section, page 3 – groups are labeled "Pkd1 ecKO" in Figure 1A, B, but aren't defined as such until later in the narrative.

As suggested, we have now simplified this terminology.

9) In many graphs, there are data points that fall outside the y-axis scales that are provided. Please correct these.

We have corrected any instances where this occurred.

10) In addition to flow, can the authors speculate on potential ligands that activate PC-1 that may be relevant to the vasculature and blood flow regulation?

We thank the reviewer for this suggestion. Some Wnt proteins have recently been described to act as PC-1 ligands (Kim et al. Nat. Cell. Biol 2016). However, we are not aware of evidence suggesting that Wnt proteins regulate arterial contractility. As such, we prefer not to speculate on this topic.

11) Does flow alters FRET between PC1/PC2.

We show that flow does not alter the size or density of PC-1 or PC-2 clusters, the PC-1 to PC-2 distance, the PC-2 to PC-1 distance, PC-1/PC-2 overlap or PC-2/PC-1 overlap. We consider this to provide sufficient evidence that flow does not alter the spatial proximity of PC-1 and PC-2 in endothelial cells.

12) In Figure 6A actin controls are not equivalent between conditions. All blots (including Figure 6A) should be quantified, normalized to controls, reported as scatter plots and statistically analyzed.

All proteins are normalized to actin in Western blotting experiments. As suggested, all blots have been quantified, normalized to controls and statistically analyzed, including those for *Pkd1/Pkd2^fl/fl^* and Pkd1/Pkd2 ecKO mouse arteries (Figure 6B). These data are now shown as a scatter plot.

13) Bottom of Page 6. Figure 6C,D should in fact refer to Figure 6B,C. Please change?

Thank you for noticing this. It has now been corrected.